# Highly Entangled 2D Ground States: Tensor Networks and Correlation Functions

Olai B. Mykland[1,2][†], Zhao Zhang[1][*]

**1** Department of Physics, University of Oslo, P.O. Box 1048 Blindern, N-0316 Oslo, Norway
**2** Department of Physics, Norwegian University of Science and Technology, NO-7491 Trondheim, Norway

† olai.b.mykland@gmail.com   * zhao.zhang@fys.uio.no

March 25, 2025

## Abstract

In this article we present analytical results on the exact tensor network representations and correlation functions of the first examples of 2D ground states with quantum phase transitions between area law and extensive entanglement entropy. The tensor networks constructed are one dimension higher than the lattices of the physical systems, allowing entangled physical degrees of freedoms to be paired with one another arbitrarily far away. Contraction rules of the internal legs are specified by a simple translationally invariant set of rules in terms of the tesselation of cubes or prisms in 3D space. The networks directly generalize the previous holographic tensor networks for 1D Fredkin and Motzkin chains. We also analyze the correlation in the spin and color sectors from the scaling of the height function of random surfaces, revealing additional characterizations of the exotic phase transitions.

# 1   Introduction

Although the average entanglement entropy (EE) in the Hilbert space of a quantum many-body system scales extensively with the subsystem size [1], for ground states (GSs) of locally interacting systems with a finite spectral gap, EE depends only on the size of the boundary between subsystems. This is known as the area law of EE [2]. Despite the name being fitting only for 3D systems, it is believed to hold in any dimension, although a rigorous proof has only been given for 1D [3]. This can be intuitively understood from the exponential decay of correlations implied by a finite energy required to excite a quasiparticle from the GS [4]. Less is known in 2D, as general theorems on the behavior of correlations and entanglement are lacking, except for a class of systems called "frustration-free", where the GS is simultaneously the common eigenstate of each local Hamiltonian term [5, 6].

When elementary excitations become massless, there is no constraint on the scaling of EE in the GS. Nevertheless, it is very rare to find even a toy model with local interactions that has beyond logarithmic scaling of EE, which are common for (1+1)D critical systems described by conformal field theories (CFTs) [7,8] and free fermionic systems with a Fermi sea in any dimension [9]. In 1D, there are basically two blueprints for designing highly entangled GSs, both with a rainbow-like distribution of entangled pairs across the center of the system. The first one is realized by an inhomogeneous interaction strength that decays from the center [10], which generalizes also to systems with Hausdorff dimension one that lives in an arbitrarily higher dimensional space [11]. The second one restores the translational invariance by introducing random entanglement patterns to the GS super-position [12–14], which can also be weighted by a deformation parameter such that the rainbow-like patterns are weighed more [15–17] in the GS. As a result of the deformation,

these models demonstrate a novel quantum phase transition characterized by an abrupt change in EE scaling.

The latter paradigm has been used to construct 2D highly entangled GSs, when arrays of 1D chains in different directions are coupled together [18, 19]. Unlike the anisotropic coupled wire models which are traditionally constructed as solvable models for topological states [20, 21], the arrays of entangled chains are coupled in an isotropic way, with the same EE scaling for bipartitions in multiple directions. The coupling at intersections, either as vertex rules or, in a less obvious way, as tesselation of 2D tiles, enables the generalization of the height function to 2D, such that each spin in the lattice knows its partner to entangle with. As a result of the coupling, or the increase in dimension, the EE scaling is different from the 1D counterparts, while there remains an entanglement phase transition from area law to extensive.

EE is closely knit with tensor networks (TNs) [22, 23], not the least because there is an easy way to read off an upper bound of the EE scaling from the geometry of the network. As efficient classical descriptions of quantum many-body states, TNs are used in many applications, not only as an important tool for tackling the major challenge of simulating quantum many body systems [24], but also bringing interesting connections to gauge-gravity dualities [25]. It is important to remember, that before TNs became a widely applicable platform for efficient numerical algorithms, in its infancy, TNs in the form of matrix product states and projected entangled pair states [23] were originally understood as exact analytic descriptions of solvable GSs, such as the valance bond solid state of the AKLT models [26]. Gradually, optimized TNs have been designed to describe states beyond the area law of EE, such as the Multiscale Entanglement Renormalization Ansatz (MERA) [24], which describes systems at quantum critical points [27], where correlations decay polynomially. These examples necessarily involve TNs that extend to one dimension higher than the physical system, and are discrete realizations of the Anti de-Sitter (AdS)/CFT correspondence [25].

Having established the far-reaching impact of studying TN descriptions of the few analytically solvable GSs, we turn to the extensively entangled examples. An exact TN representation of the GSs of the 1D Motzkin and Fredkin chains was found in Ref. [28]. The 2D bulk of the TN allows entangled physical degrees of freedom on its boundary to be connected to each other and each non-vanishing entry of the tensor product of the network corresponds to a unique way of pairing the physical spins. Unlike the MERA, which readily generalizes to 2D [29, 30], this prescription faces fundamental challenges when one attempts to generalize to the 2D highly entangled GSs of Ref. [18, 19]. There, the spins are entangled in multiple directions, and the flows connecting entangled pairs in the 3D network should be both independent, recovering the 2D TN when viewed as cross-sections, and coupled, when considered as a 3D whole. Furthermore, its vertex rules that couple the physical chains need to be somehow intrinsically enforced in the bulk of the TN, by the contraction of internal legs. Both of these features are naturally manifested in our findings, which is the first example of an exact TN representation of an extensively entangled GS of a 2D system.

The rest of the article is structured as follows. We begin with a brief review of the GSs of the Fredkin chain and its 2D generalizations in Sec. 2. In Sec. 3 and 4, we present the TN representations of the GSs for the two 2D generalizations, based on colored 6-vertex model and lozenge tiling respectively. Sec. 5 studies the behaviors of both the spin and color correlations for the 1D and 2D highly entangled GSs, in the scaling limit. We conclude the article, with a discussion of possible future work in Sec. 6.

## 2 Review of highly entangled GSs

### 2.1 Single 1D Fredkin chains

The Fredkin spin chains [13,14] are 1D models of interacting half-integer spins, introduced as complements to the integer spin models, known as the Motzkin chains [12, 31]. The GSs of both models exhibit logarithmic (resp. power law) scaling of entanglement entropy for spin-$\frac{1}{2}$ or spin-1 (resp. higher spins). In [16], the GS was deformed by a deformation parameter $q$ to exhibit a phase of extensive entanglement entropy. The Hamiltonian is written as a sum of projection operators, and is presented in Appendix A.1. The GS of the model can be described as a superposition of chromatically correlated Dyck paths. An $N$ step colored Dyck path is defined as a path on the $(x, y)$ grid, taking up steps $(1,1)^c$ and down steps $(1, -1)^c$ of color $c$ starting from $(-1/2, 0)$ and ending at $(N - 1/2, 0)$, that never passes below the $x$-axis. Clearly, $N$ must be even. A chromatically correlated Dyck path is then defined as a colored Dyck path, where the color of a down step matches that of the closest unmatched up step to the left. Colored spin configurations can be represented by colored Dyck paths on the grid by making the correspondence between a spin at position $i$ in the chain and a step taken at position $i - 1/2$ in walk in the following way

$$| \uparrow^c \rangle_i \leftrightarrow (1,1)^c_{i-1/2} \text{ and } | \downarrow^c \rangle_i \leftrightarrow (1, -1)^c_{i-1/2}. \tag{2.1}$$

The subscript $i - 1/2$ indicates the midpoint between spins. Using this correspondence, the unique GS of the colorful deformed Fredkin spin chain $|\text{GS}_{\text{Fredkin}}(q)\rangle$, can be expressed as

$$|\text{GS}_{\text{Fredkin}}(q)\rangle = \frac{1}{\mathcal{N}_{\text{Fredkin}}} \sum_{w \in D_N^s} q^{A(w)} |w\rangle, \tag{2.2}$$

where $D_N^s$ is the set of all color correlated Dyck walks on $N$ steps where each step can take on $s$ different colors, $A(w)$ is the total area beneath a Dyck walk $w$ and $\mathcal{N}_{\text{Fredkin}}$ is a normalization constant. In terms of a height function $\phi_i$, defined as the height of the Dyck walk at $i$, each spin configuration $|w\rangle$ in Eq. (2.2) is described by $\phi_i \geq 0$ for all $i$. This is the characteristic property generalized to the 2D models presented later. The uniqueness of the GS, and its extensive entanglement entropy for $q > 1$, were shown in [16]. As pointed out in [17], the volume scaling of entanglement entropy is due to the color degree of freedom.

### 2.2 Two arrays of Fredkin chains coupled by 6-vertex rules

A 2D generalization of the single colorful deformed Fredkin chain is realized by overlaying orthogonal arrays of them coupled at their intersections by the ice rule [18]. The model Hamiltonian is presented in Appendix A.2. The spins of the model live on the edges between vertices in a square lattice, as seen in panel (a) of Fig. (1), and we divide the spins into horizontal and vertical spins, denoted $S^{\text{h}}_{x,y}$ and $S^{\text{v}}_{x,y}$ respectively, see panel (b) of Fig. (1). The horizontal (resp. vertical) spins is in one of two spin states: up (resp. left) or down (resp. right). In addition, we let each spin take on two different colors, red or blue. A horizontal spin corresponds to a vertical edge in the lattice and a vertical spin corresponds to a horizontal edge. The reason behind this seemingly confusing naming convention, is that the horizontal (resp. vertical) spins participate in a horizontal (resp. vertical) Fredkin spin chain. A horizontal spin chain, is a spin chain consisting of horizontal spins $S^{\text{h}}_{x,y}$ at a given $y = a$ and a vertical spin chain is a spin chain consisting of vertical spins $S^{\text{v}}_{x,y}$ at a given $x = a$. The horizontal Fredkin chain at $y = 1$ and the vertical Fredkin chain at

$x = 3$ are marked by thick spins in panel (a) in Fig. (1), and their spin configurations are described by the Dyck walks below and to the right of the 2D spin system.

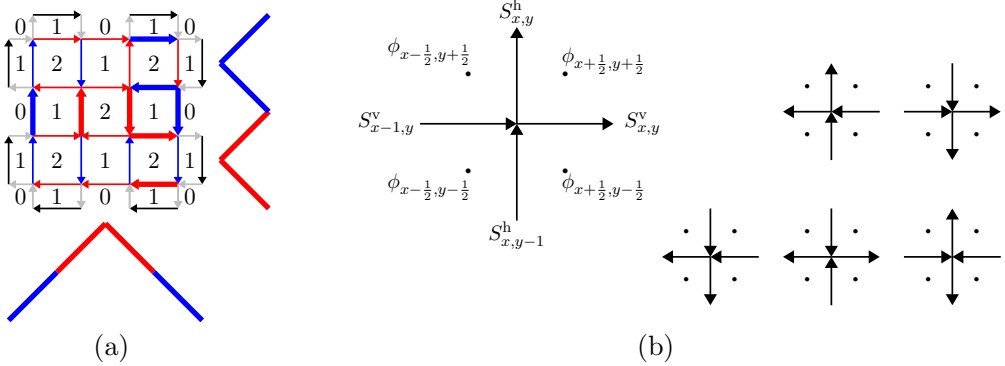

Figure 1: (a) Maximal volume spin configuration present in Eq. (2.5) for the $L = 4$ system. The numbers indicate the height $\phi$ on the dual lattice. The black and gray spins are spins outside our system implicitly fixed by the boundary conditions. The Dyck walks in $y = 1$ and $x = 3$ horizontal and vertical spin chains are shown below and to the right. (b) The 6 different vertex configurations compatible with the ice-rule. Horizontal $S^{\mathrm{h}}$ and vertical $S^{\mathrm{v}}$ spins are indicated along with the height function $\phi$ defined at the dual lattice points, marked with the •'s.

Adjacent spin chains are coupled through an ice rule term, penalizing spin configurations that are not compatible with the ice-rule. Compatible spin configurations are those where there is an equal number of spins pointing towards and away from the vertex, and are seen in panel (b) in Fig. (1). Importantly, the ice rule enables a well defined height function $\phi_{x+1/2,y+1/2}$ on the vertices of the dual lattice, marked with black dots in Fig. (1) (b). The height change convention is defined as

$$
\begin{aligned}
\phi_{x-\frac{1}{2},y+\frac{1}{2}} &= \phi_{x-\frac{1}{2},y-\frac{1}{2}} + 2S^{\mathrm{v}}_{x-1,y}, \\
\phi_{x+\frac{1}{2},y+\frac{1}{2}} &= \phi_{x+\frac{1}{2},y-\frac{1}{2}} + 2S^{\mathrm{v}}_{x,y}, \\
\phi_{x+\frac{1}{2},y-\frac{1}{2}} &= \phi_{x-\frac{1}{2},y-\frac{1}{2}} + 2S^{\mathrm{h}}_{x-1,y}, \\
\phi_{x+\frac{1}{2},y+\frac{1}{2}} &= \phi_{x-\frac{1}{2},y+\frac{1}{2}} + 2S^{\mathrm{h}}_{x,y}.
\end{aligned}
\tag{2.3}
$$

The unique GS of the 6-vertex model is given by a superposition over a set of spin configurations $S_{L \times L}$ on the $L \times L$ lattice, where each $S \in S_{L \times L}$ satisfies the ice rule at all ice rule vertices. In addition, each spin configuration $S \in S_{L \times L}$ in the GS is described by color correlated Dyck walks in all horizontal and vertical spin chains. This means that for the spin configurations $S \in S_{L \times L}$, we have $\phi_{x+1/2,y+1/2} \geq 0$ for all $x, y$. In the GS superposition, spin configurations are weighted by the volume $V$ of a spin configuration, defined as

$$
V = \sum_{x,y=0,0}^{L-2} \phi_{x+\frac{1}{2},y+\frac{1}{2}}.
\tag{2.4}
$$

The GS of the 6-vertex model can then be written as

$$
|\mathrm{GS}_{6-\mathrm{vertex}}(q)\rangle = \frac{1}{\mathcal{N}_{6-\mathrm{vertex}}} \sum_{S \in S_{L \times L}} {\sum_{C}}' q^{V(S)} |S^C\rangle.
\tag{2.5}
$$

The primed sum over colorings $C$ indicates that we only sum over colorings giving color correlated Dyck walks in the horizontal and vertical spin chains. The maximal volume spin configuration present in Eq. (2.5) for the $L = 4$ system is shown in panel (a) in Fig. (1). Below (resp. to the right of) the spin configuration, the color correlated Dyck walk in the $y = 1$ (resp. $x = 3$) horizontal (resp. vertical) spin chain is shown. For large values of $q$ the high volume spin configurations will dominate the superposition in Eq. (2.5). This will result in long range color correlations, just as as the case was in Eq. (2.2).

## 2.3   Three arrays of Fredkin chains coupled by lozenge tiling rules

Another 2D generalization of the single colorful Fredkin chain is realized by dimer covering of the honeycomb lattice, or equivalently, the tiling of lozenges in three different orientations on the triangular lattice, where the dimers or lozenges are polychromatic [19]. This generalization was inspired by the monochromatic 2D generalization proposed by Ref. [32], and the model Hamiltonian is presented in Appendix A.3.

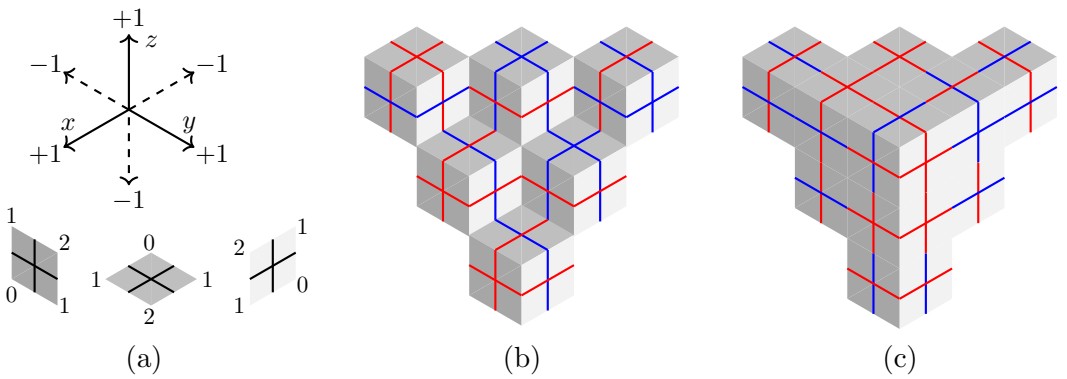

Figure 2: (a) Upper panel shows coordinate system and height convention. Lower panel shows the three different lozenges, where the numbers indicate how the height change around a lozenge according to the height convention. (b)-(c) Two color decorated lozenge tilings for a given boundary. The minimal volume tiling is seen in (b), while the maximal volume tiling is seen in (c). The three-dimensional effect is understood by noting that the light is shining onto the lozenge tilings anti-parallel with the $y$-axis, as defined in (a). In both lozenge tilings we have color correlated Dyck walks, in the $xy$-, $xz$- and $yz$-plane.

In our context, the important property of lozenge tilings is that they enable a well defined height function $\phi$ at the vertices $u$ of the triangular lattice [33]. The height changes if we move along the edge of a lozenge, according to the height convention defined in panel (a) in Fig. (2). It is also useful to define the height of an edge $u - v$ connecting two height vertices $u$ and $v$, as the average height of $u$ and $v$. We can then define the volume of a given lozenge tiling $T$ as the sum of the height at the height vertices $u$ or the edges $u - v$.

$$V(T) = \sum_{u \in \mathcal{R}} \phi_u = \sum_{u - v \in \mathcal{R}} \phi_{u-v}, \tag{2.6}$$

where the sum is over the domain $\mathcal{R}$. Examples of minimal and maximal volume lozenge tilings on a domain $\mathcal{R}$, can be seen in panel (b) and (c) in Fig. (2). In these examples we have defined the height on the boundary height vertices to be alternating between $\pm 1/2$, which means that the height on the boundary edges is 0. Note that in the tilings in Fig. (2) we have color correlated Dyck walks, in the $xy$-, $xz$- and $yz$-plane. By following a path

of colored lines, one is moving in either of the three planes defined by the axes in panel (a) Fig. (2) and a color correlated Dyck walk is traced out. The three-dimensional effect in these figures is understood by noting that the light is shining onto the lozenge tiling anti-parallel with the $y$-axis. Finally, we point out that in order for a domain $\mathcal{R}$ of the triangular lattice $\Lambda$ to be tileable by lozenges, one must be able to define a height function $\phi$ on the height vertices $u$ on the boundary $\partial\mathcal{R}$ that satisfies two requirements [34]. Firstly, for all edges $u - v$ on the boundary, the height change between $u$ and $v$ in the positive direction must be exactly 1. Secondly, we must have

$$\forall u, v \in \partial\mathcal{R}: \quad |\phi_u - \phi_v| \leq d(u, v), \tag{2.7}$$

where $d(u, v)$ is the minimal number of steps taken in positive directions connecting $u$ and $v$. The domain seen in panel (b) and (c) in Fig. (2) satisfies a stronger constraint, where $d(u, v)$ is replaced by 1. As we will see in Appendix D, this implies that the domain is tileable by hexagons. As in the previous models, the GS of the quantum lozenge tiling model is given by the state that is annihilated by each term in the model Hamiltonian. The terms in the quantum lozenge tiling Hamiltonian are constructed so that the height function $\phi$ for lozenge tilings in the GS superposition is constrained to take on positive values at the edges $u - v$. The lozenge tilings in Fig. (2), are examples of such lozenge tilings. In addition, we have a superposition over color correlation in the three planes. The GS can be written as

$$|\text{GS}_{\text{lozenge}}(q)\rangle = \frac{1}{\mathcal{N}_{\text{lozenge}}} \sum_{T \in T_{\mathcal{R}}} {\sum_{C}}' q^{V(T)} |T^C\rangle, \tag{2.8}$$

where the sum is over lozenge tilings $T$ from the set of lozenge tilings $T_{\mathcal{R}}$ on the domain $\mathcal{R}$, where the $\phi_{u-v} \geq 0$ for $u - v \in \mathcal{R}$. For a domain where the boundary edges have height 0, this implies that these lozenge tilings are described by Dyck walks in all three planes. The primed sum over the coloring $C$ as usual indicates that we only sum over colorings giving color correlated Dyck walks. Just as in the previous GS's, for large values of $q$ the high volume spin configurations will dominate the superposition which will result in long range color correlations, giving rise to the unusual entanglement properties.

## 3 TN for the colored 6-vertex model

In this section, we find an exact TN representation of the GS of the 6-vertex coupled Fredkin chains using a tiling method. In this tiling method, one aims to construct a set of tiles whose possible tilings has a one-to-one correspondence with individual physical configurations in the GS. One obtains the TN representation by letting each tile define a non-vanishing entry of the constituent tensor of the TN. The general procedure is explained in Sec. 3.1 by reviewing the construction of the TN for 1D GS, and pointing out the additional ingredients necessary for the couplings at vertices in the 2D counterparts. Such a method does of course not pin down an unique form of tensors, as expected for any TN due to reasons like gauge degree of freedom. Our findings of a working choice of the tensor, is symmetric and closely mimics the 1D case, and is presented in Sec. 3.2. In Appendices. B.1 and B.2 we use the height function to show that the choice of tiles for the 6-vertex model ensure a one-to-one correspondence between possible tilings and spin configurations in the GS. Finally, in Sec. 3.3, we specify explicitly the nonzero entries of the tensors with incoming and outgoing indices, for the $q$-deformed model, making a connection to U(1) invariant TNs studied in [35].

## 3.1 TN from tiling

To construct exact TN representations of the GSs of the 2D models, we employ the same method used in Ref. [28] to obtain an exact holographic TN representation of the GS of the single Fredkin spin chain. In this method the correspondence between spin configurations and walks is used to construct the TN representation. This is achieved through an intermediate step where a valid tiling of a set of tiles is associated to each walk present in the GS. In this context, a valid tiling is a tiling where each edge of each tile matches its adjacent tiles. By carefully choosing an appropriate set of tiles, one can assure a one-to-one correspondence between valid tilings and spin configurations present in the GS superposition.

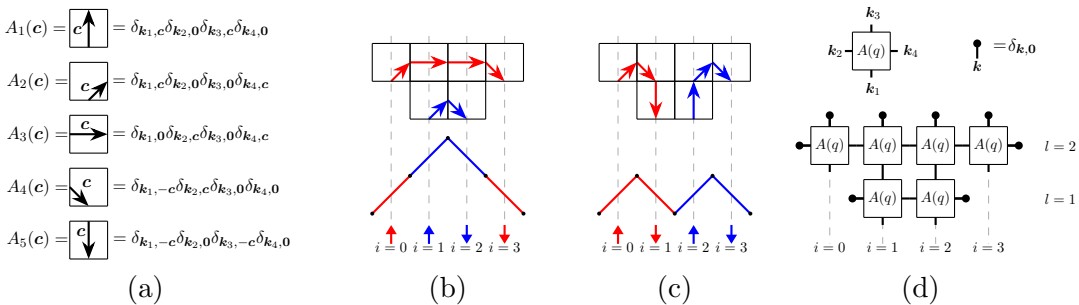

Figure 3: (a) The five different tiles $A_i(\boldsymbol{c})$ for the single Fredkin chain case. The tiles are also defined as rank-4 tensors in terms of Kronecker deltas, where the indices $\boldsymbol{k}_i$ are defined as for $A(q)$ in panel (d). We have $\boldsymbol{c} = (1,0)$ for red arrow, $\boldsymbol{c} = (0,1)$ for blue arrow. No arrow corresponds to $\boldsymbol{0}$. (b)-(c) Valid tilings corresponding to the maximal and minimal height Dyck walk for the $L = 4$ system. (d) TN representation of the GS of the single Fredkin chain for $L = 4$. The constituent tensors are defined in the upper panel. The variable $l$ denotes the different levels of the holographic TN.

For spin configurations present in the GS of the Fredkin model, which are described by Dyck walks, this one-to-one correspondence can be assured by the set of tiles $A_i(\boldsymbol{c})$ seen in panel (a) of Fig. (3). In panel (b) and (c) we see that the valid tilings contain arrowed paths indicating the color correlation between spins in the given Dyck walk. A colored spin up (resp. down) at $i$ is represented by a colored arrowed path entering (resp. leaving) the tiling at the floor edge of the bottom tile in the tower of tiles at $i$. Clearly, each valid tiling corresponds to a unique way of pairing the degrees of freedom. This set of tiles enables exactly one valid tiling for any given Dyck walk and no valid tiling for non-Dyck walks. The tensor building up the TN representation is then defined by the set of tiles in the following way: The number of indices of the tensor is determined by the shape of the tiles and each tile represents an index configuration of the tensor giving a nonzero value. Contractions between two tensors in the TN, then correspond to summing over tile configurations matching at that edge. By weighing each tile with an appropriate value of the deformation parameter $q$, one can ensure that contracting the TN yields the weighted superposition of spin configurations in the GS. In the single Fredkin chain case, this method gives a rank-4 tensor $A(q)$ as

$$A(q) = \sum_{\boldsymbol{c}=(1,0),(0,1)} \left[ A_1(\boldsymbol{c}) + \sqrt{q} A_2(\boldsymbol{c}) + q A_3(\boldsymbol{c}) + \sqrt{q} A_4(\boldsymbol{c}) + A_5(\boldsymbol{c}) \right]. \tag{3.1}$$

The indices $\boldsymbol{k}_1, \cdots, \boldsymbol{k}_4$ of $A(q)$ are defined in panel (d) of Fig. (3) and have been suppressed in Eq. (3.1). We use a vector notation for the indices to keep track of the color of

the arrows, as this notation is convenient to extend to the tensors in the TN for the 2D systems. It is clear that one can simply replace the different vectors with a unique number, to arrive at the more familiar way of denoting index configurations by numbers. Note also that we use a minus sign to indicate arrows leaving (resp. coming in) at the $\boldsymbol{k}_1$ (resp. $\boldsymbol{k}_3$) index. This is also conveniently extended to the 2D cases, and is discussed in more detail in section 3.3. In Eq. (3.1), we consider two colors, namely red, $\boldsymbol{c} = (1, 0)$, and blue, $\boldsymbol{c} = (0, 1)$. Note that each of the tiles $A_i(\boldsymbol{c})$ can be defined as a rank-4 tensor expressed as a product of Kronecker deltas, as seen in panel (a) in Fig. (3). The shape of the resulting TN naturally becomes that of the valid tilings, giving an inverse step pyramid shape for the GS of the Fredkin model as seen in panel (d) in Fig. (3). Note that the boundary tensors $\delta_{\boldsymbol{k},\boldsymbol{0}}$ ensures that "no arrows flow out" of the TN, except at the bottom edges.

Before fixing the constituent tensors in the next subsection, we can already determine the geometry of the TN from the EE scaling, which is upper-bounded by the zeroth Rényi entropy. If we use a translationally invariant definition of individual tensors, the bond dimension of each internal leg is a constant. The latter then becomes proportional to the minimal number of bonds traversing the boundary of a bipartition, which means the vertical cross-sectional area of the network should grow quadratically with the linear dimension of the subsystem. For the square lattice model of couple Fredkin chains, this means a 3D geometry like an (upside-down) pyramid.

## 3.2  Cubic tiles

To arrive at an appropriate set of tiles, we first require that each physical spin corresponds to a single tensor at an external leg of the TN, as in the 2D TN in Fig. (3). Then the geometry of the physical lattice dictates that the tensors should be arranged as shown in Fig. (4) (a). We also adopt the holographic geometry of the 2D TN, which enables correlations between physical spins to be propagated in various planes above the physical lattice. Then each square in Fig. (4) (a) is the projection of a tower of cubes corresponding the same physical spin, implying that all of the cubes in a given tower are of the same type.

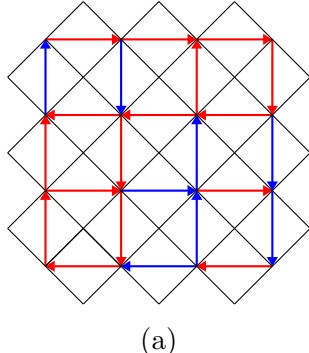
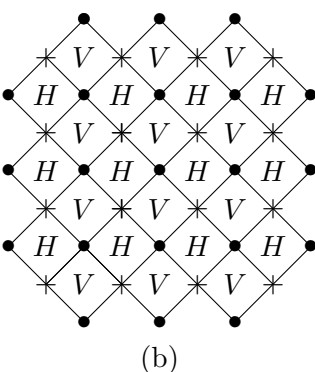

(a)  (b)

Figure 4: (a) Top view of towers of cubical tiles associated to each spin, for a $L = 4$ system. (b) The two sublattices corresponding to horizontal tiles $H$ and vertical tiles $V$. The original lattice vertices are located at +'s and height functions are defined on the dual lattice at ●'s.

From the previous considerations on the EE scaling and the TN geometry, we infer that the towers must be taller in the center of the lattice and shorter near the boundary. As shown in Fig. (4) (b), this tensor arrangement defines two sublattices: one for the

horizontal spins with tensors $H$, and one for the vertical spins with tensors $V$. Within each level of the TN, the tensors contract only with those that belong to the other sublattice. Before constructing the appropriate cubic tiles, one can note that a TN representation of the GS of the colorless 6-vertex model, or fully-packed loop model, is given in terms of a PEPS with quadrivalent tensors [36]. In this PEPS the physical degrees of freedoms live on indices contracted between the square tensors. In our case, where the physical degrees of freedom are enlarged with color, we instead associate a tensor to each physical degree of freedom and promote the tensors to cubes. Such an extrusion from 2D to 3D enables entangled pairs farther away more straightforwardly to be connected in the TN representation.

In order to construct an appropriate set of cubic tiles, we now consider the constraints on the spin configurations in the GS Eq. (2.5). The spin configurations are subject to constraints in three different planes: the ice-rule constraint in the $xy$-plane of the lattice, and the color correlated Dyck walk rule in all of the $xz$- and $yz$-planes, corresponding to each individual spin chain in Fig. (1). We know that the square tiles in Fig. (3) incorporate the color correlated Dyck walk rule for the single Fredkin chain, enabling valid tilings only for color correlated Dyck walks. We therefore aim to construct cubic tiles that closely mimic the square tiles for the single Fredkin chain, to incorporate the Dyck walk rule for each horizontal and vertical spin chain. Since we have Dyck walks in both horizontal and vertical spin chains for spin configurations in the 2D GS, we depict arrows due to horizontal spins with dashed lines and arrows due to vertical spins with solid lines. Then, we ensure the Dyck rule in both horizontal and vertical spin chains, by requiring arrow continuity at all faces for both dashed and solid arrows.

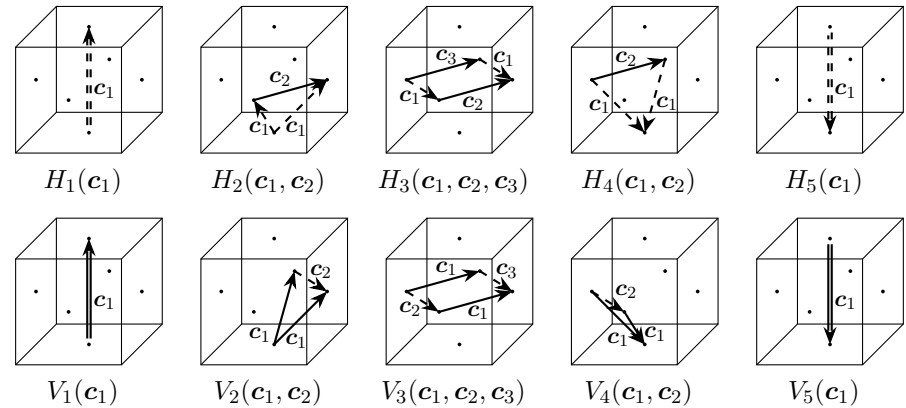

$$H_1(\boldsymbol{c}_1) \qquad H_2(\boldsymbol{c}_1, \boldsymbol{c}_2) \qquad H_3(\boldsymbol{c}_1, \boldsymbol{c}_2, \boldsymbol{c}_3) \qquad H_4(\boldsymbol{c}_1, \boldsymbol{c}_2) \qquad H_5(\boldsymbol{c}_1)$$

$$V_1(\boldsymbol{c}_1) \qquad V_2(\boldsymbol{c}_1, \boldsymbol{c}_2) \qquad V_3(\boldsymbol{c}_1, \boldsymbol{c}_2, \boldsymbol{c}_3) \qquad V_4(\boldsymbol{c}_1, \boldsymbol{c}_2) \qquad V_5(\boldsymbol{c}_1)$$

Figure 5: The set of cubical tiles corresponding to nonzero entries of a tensor in the network. The dashed (resp. solid) arrows are due to horizontal (resp. vertical) physical spins on the 2D lattice. Arrows pointing between top and bottom surfaces of a cube are doubled, and share the same color, as they come from the same spin. The labels $\boldsymbol{c}_i$ denote the colors of the arrows.

To incorporate the Dyck rule in the cubic tiles, we require that horizontal (resp. vertical) cross-sections in the $xz$-(resp. $yz$-)plane of valid cube tilings reproduce the tilings in the 2D case, seen in Fig. (3). This requirement fixes the configuration of dashed (resp. solid) arrows in the $H$ (resp. $V$) tiles, to similar arrow configurations as in the square tiles. This means that the dashed (resp. solid) arrows in the $H$ (resp. $V$) tiles either go between the floor and the roof (as in $A_1$ and $A_5$), between the floor and the walls (as in $A_2$ and $A_4$) or between opposite walls (as in $A_3$). This also means that arrows corresponding to each type of spin would necessarily traverse cubes in both sublattices, as shown in Fig. (7) (b) and (c). The requirement of continuity of arrows of each type,

ensures that the dashed (resp. solid) arrows always travel along the $x$-(resp. $y$-)direction in the $V$ (resp. $H$) tiles. However, the requirement put on the cross-sections of the cube tilings, enables two different configurations of the dashed (resp. solid) arrows for the cube tiles $H_2$ and $H_4$ (resp. $V_2$ and $V_4$) corresponding to the $A_2$ and $A_4$ square tiles. These two correspond to the two different walls of the cube, the arrow from (resp. to) the floor can go to (resp. from.) As it turns out, by including arrows from both configurations, that is using two arrows of the same type per spin, the ice-rule will also be enforced in the tilings through arrow continuity. This choice of including arrows from both configurations is seen in the cube tiles in Fig. (5). The fact that the ice-rule can be enforced through arrow-continuity by using two of the same type per spin, is established in Appendix B.2, but can be understood intuitively by noting that each spin is interacting with other spins through the ice-rule at two vertices. Therefore, it is reasonable that the value of a spin must be represented by two arrows in order for the spin value to be "communicated" to spins at two different vertices. In Appendix B, it is shown that there is an one-to-one correspondence between spin configurations $|S^C\rangle$ in the GS Eq. (2.5) and valid tilings of the cube tiles in Fig. (5).

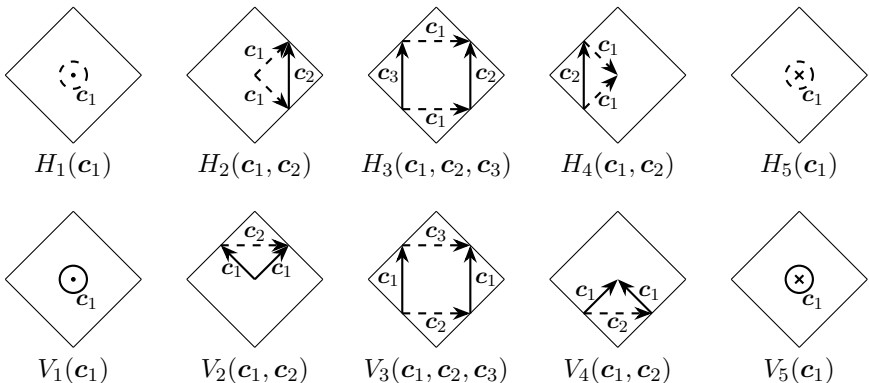

$$H_1(\boldsymbol{c}_1) \qquad H_2(\boldsymbol{c}_1, \boldsymbol{c}_2) \qquad H_3(\boldsymbol{c}_1, \boldsymbol{c}_2, \boldsymbol{c}_3) \qquad H_4(\boldsymbol{c}_1, \boldsymbol{c}_2) \qquad H_5(\boldsymbol{c}_1)$$

$$V_1(\boldsymbol{c}_1) \qquad V_2(\boldsymbol{c}_1, \boldsymbol{c}_2) \qquad V_3(\boldsymbol{c}_1, \boldsymbol{c}_2, \boldsymbol{c}_3) \qquad V_4(\boldsymbol{c}_1, \boldsymbol{c}_2) \qquad V_5(\boldsymbol{c}_1)$$

Figure 6: Top view of the cubical tiles in Fig. (5).

For convenience we introduce a way of drawing the cube tiles in Fig. (5) as square tiles in Fig. (6), which can be understood as top views of the cube tiles. The symbols ⊙ and ⊙ represent two dashed and solid arrows, respectively, pointing out of the plane of the paper. Similarly, ⊗ and ⊗ denote two dashed and solid arrows, respectively, pointing into the plane of the paper. The arrows going between the center of a square and one of the edges of the square correspond to arrows going between the bottom of a cube and one of the walls of the cube.

To set up for showing the one-to-one correspondence between physical configurations in the GS and valid tilings, which is carried out in Appendix. B, we give two illustrative examples of tilings for the spin configurations of minimal and maximal height with an $L = 4$ system in Fig. (7). The top view of the tilings are given for the two levels $l = 1$ (panel (b)) and $l = 2$ (panel (c)) separately. The 3D tilings are understood to consist of the $l = 2$ level tiles placed on top of the $l = 1$ level, such that the asterisks $*$ overlap. The black arrows seen in the $l = 2$ level represent the influence of the black spins outside our system, fixing the height of even numbered spin chains to one unit higher than odd spin chains. They are ultimately a consequence of choosing a square shaped boundary of the physical system, which results in the four corners being frozen outside the 'arctic circle' of the six-vertex models. We see from the cutouts in Fig. (8), which are marked by thick black lines in Fig. (7), that cube tilings of horizontal and vertical spin chains, clearly correspond to the 2D tilings in Fig. (3).

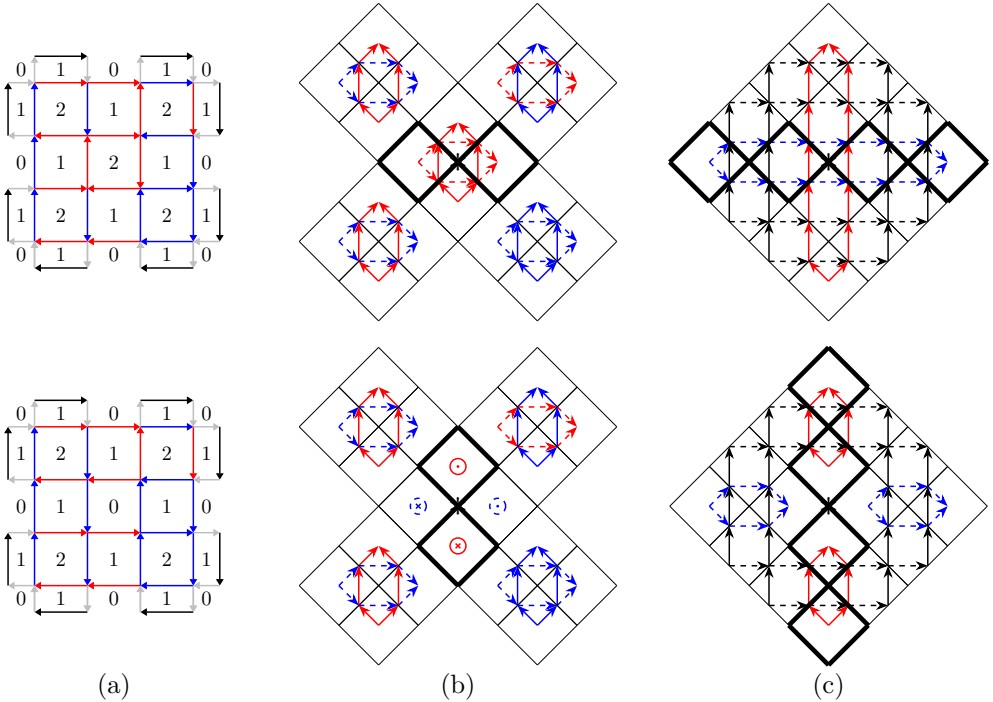

Figure 7: The highest (upper panels) and lowest (lower panels) height configurations with a certain coloring for an $L = 4$ square lattice system: (a) The physical spin and height configurations (along with the fictitious spins outside the boundary in gray and black). (b) The tilings in the $l = 1$ level. (c) The tilings in the $l = 2$ level, where the black arrows are due to the black spins outside the system. The asterisk $*$ is a visual aid marking the center of the tilings, corresponding to the center of the central square in (a). The thick black lines mark cutouts of the tiling, which can be seen in Fig. (8)

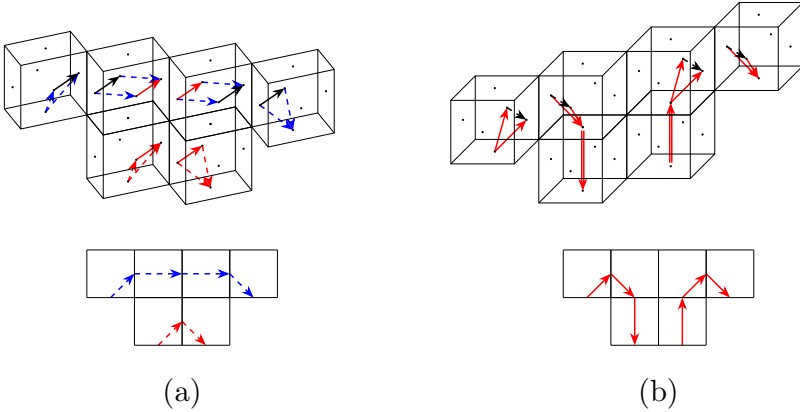

Figure 8: Upper panels show cutouts of the central horizontal and vertical spin chains from the valid tiling corresponding to the highest and lowest height configurations, highlighted with thick black lines in Fig. (7), shown in (a) and (b), respectively. The lower panel shows the corresponding tilings for the single chain.

### 3.3 From cubic tiles to 6-leg tensors

We now define the cube tensors in the same way the square tensor $A(q)$ was defined in Sec. 3.1. We denote tensors in the sublattice for horizontal (resp. vertical) spins with $H(q)$ (resp. $V(q)$). Their indices are labeled $k_1, \ldots, k_6$ as specified in Fig. (9). In Fig. (9), panel (a) shows how each face of the cube corresponds to an index and panel (b) shows this for the square drawing of the tensor. Note that indexing is the same for both $H(q)$ and $V(q)$ tensors. So the $k_2$ leg of one $H(q)$ tensor will contract with the $k_4$ leg of one adjacent $V(q)$ tensor, while its $k_3$ leg will be contracted with the $k_5$ leg of another $V(q)$ tensor. The $k_1$ and $k_6$ legs are contracted between the tensors in the same tower.

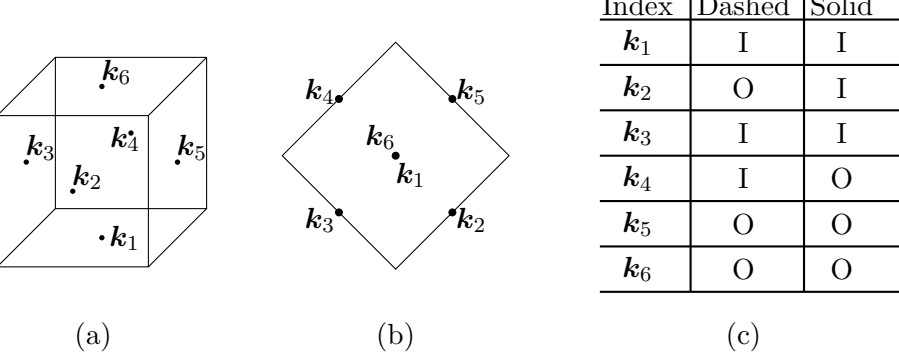

| Index | Dashed | Solid |
|:-:|:-:|:-:|
| $k_1$ | I | I |
| $k_2$ | O | I |
| $k_3$ | I | I |
| $k_4$ | I | O |
| $k_5$ | O | O |
| $k_6$ | O | O |

(a)            (b)            (c)

Figure 9: (a)-(b) Depiction of the indices $k_1, ..., k_6$ and the corresponding cube faces, where (b) indicates this in the square drawings of the cube tiles. Note that the $k_1$ and $k_6$ indices coincide in the square drawings, as they are top views of the cube tiles. (c) Classification of the indices in terms of incoming (I) and outgoing (O) with respect to dashed and solid arrows.

For each cubical tile, the number of arrows going into the cube tile equals the number of arrows leaving the cube tile for both arrow types. The arrowed notation, along with the continuity of arrows for each cube tile, suggest a connection with the U(1) invariant tensors discussed in Ref. [35] and considered in Ref. [37] in connection with an exact holographic TN representation of the GS of the Motzkin chain. In Ref. [35], tensors are interpreted as linear maps between the vector spaces associated to its incoming indices and the vector space associated to its outgoing indices. In our case, we define an index as outgoing (resp. incoming) for a given arrow type, if arrows of that type flow out of (resp. into) the cube at the corresponding face in all the cube tiles. This makes $k_3$, $k_4$ the incoming for dashed lines, and $k_2$, $k_3$ the incoming for solid lines. Defining indices as incoming and outgoing in this way, limits the range of possible values for each index. We also define $k_1$ as an incoming index and $k_6$ as an outgoing index for both arrow types. The classification of the indices in terms of incoming (I) and outgoing (O) for the two arrow types is summarized in Fig. (9) (c). Note that the contractions between indices, such as $k_3$ and $k_5$, are always between one incoming and one outgoing for a given arrow type.

Since each tensor leg can have two types of arrows, solid and dashed, in two different colors, it can be specified by a 4-component vector, with each component denoting respectively: red solid, blue solid, red dashed, and blue dashed arrows. A component being $\mathbf{0} = (0,0)$ means no arrow of that type, of either color, goes into or out of the cubical tile along that leg. A component being 1 means there is one arrow in a certain color and of a certain type goes into the cube, if the leg is an incoming index, or out of the cube if the index is outgoing. A component being two means two arrows of a given type. Negative values mean that the arrows go in the opposite direction. Hence each leg is assigned a

vector value $\boldsymbol{k}_i = (\boldsymbol{c}_{\text{dashed}}, \boldsymbol{c}_{\text{solid}})$, where $\boldsymbol{c}_{\text{dashed(solid)}} = (1,0)$ or $(0,1)$ for red or blue arrows. With this notation for the tensor indices, the non-vanishing entries of the tensors are summarized in Table. 1.

Table 1: Nonzero entries of the tensor $H(q)$ (top table) and $V(q)$ (bottom table) corresponding to tiles $H_i$ and $V_i$, and their corresponding weight.

| Tile | $\boldsymbol{k}_1$ | $\boldsymbol{k}_2$ | $\boldsymbol{k}_3$ | $\boldsymbol{k}_4$ | $\boldsymbol{k}_5$ | $\boldsymbol{k}_6$ | Weight |
|---|---|---|---|---|---|---|---|
| $H_1$ | $(2\boldsymbol{c}_1, \boldsymbol{0})$ | $(\boldsymbol{0}, \boldsymbol{0})$ | $(\boldsymbol{0}, \boldsymbol{0})$ | $(\boldsymbol{0}, \boldsymbol{0})$ | $(\boldsymbol{0}, \boldsymbol{0})$ | $(2\boldsymbol{c}_1, \boldsymbol{0})$ | $1$ |
| $H_2$ | $(2\boldsymbol{c}_1, \boldsymbol{0})$ | $(\boldsymbol{c}_1, \boldsymbol{c}_2)$ | $(\boldsymbol{0}, \boldsymbol{0})$ | $(\boldsymbol{0}, \boldsymbol{0})$ | $(\boldsymbol{c}_1, \boldsymbol{c}_2)$ | $(\boldsymbol{0}, \boldsymbol{0})$ | $q^{1/4}$ |
| $H_3$ | $(\boldsymbol{0}, \boldsymbol{0})$ | $(\boldsymbol{c}_1, \boldsymbol{c}_2)$ | $(\boldsymbol{c}_1, \boldsymbol{c}_3)$ | $(\boldsymbol{c}_1, \boldsymbol{c}_3)$ | $(\boldsymbol{c}_1, \boldsymbol{c}_2)$ | $(\boldsymbol{0}, \boldsymbol{0})$ | $\sqrt{q}$ |
| $H_4$ | $(-2\boldsymbol{c}_1, \boldsymbol{0})$ | $(\boldsymbol{0}, \boldsymbol{0})$ | $(\boldsymbol{c}_1, \boldsymbol{c}_2)$ | $(\boldsymbol{c}_1, \boldsymbol{c}_2)$ | $(\boldsymbol{0}, \boldsymbol{0})$ | $(\boldsymbol{0}, \boldsymbol{0})$ | $q^{1/4}$ |
| $H_5$ | $(-2\boldsymbol{c}_1, \boldsymbol{0})$ | $(\boldsymbol{0}, \boldsymbol{0})$ | $(\boldsymbol{0}, \boldsymbol{0})$ | $(\boldsymbol{0}, \boldsymbol{0})$ | $(\boldsymbol{0}, \boldsymbol{0})$ | $(-2\boldsymbol{c}_1, \boldsymbol{0})$ | $1$ |

| Tile | $\boldsymbol{k}_1$ | $\boldsymbol{k}_2$ | $\boldsymbol{k}_3$ | $\boldsymbol{k}_4$ | $\boldsymbol{k}_5$ | $\boldsymbol{k}_6$ | Weight |
|---|---|---|---|---|---|---|---|
| $V_1$ | $(\boldsymbol{0}, 2\boldsymbol{c}_1)$ | $(\boldsymbol{0}, \boldsymbol{0})$ | $(\boldsymbol{0}, \boldsymbol{0})$ | $(\boldsymbol{0}, \boldsymbol{0})$ | $(\boldsymbol{0}, \boldsymbol{0})$ | $(\boldsymbol{0}, 2\boldsymbol{c}_1)$ | $1$ |
| $V_2$ | $(\boldsymbol{0}, 2\boldsymbol{c}_1)$ | $(\boldsymbol{0}, \boldsymbol{0})$ | $(\boldsymbol{0}, \boldsymbol{0})$ | $(\boldsymbol{c}_2, \boldsymbol{c}_1)$ | $(\boldsymbol{c}_2, \boldsymbol{c}_1)$ | $(\boldsymbol{0}, \boldsymbol{0})$ | $q^{1/4}$ |
| $V_3$ | $(\boldsymbol{0}, \boldsymbol{0})$ | $(\boldsymbol{c}_2, \boldsymbol{c}_1)$ | $(\boldsymbol{c}_2, \boldsymbol{c}_1)$ | $(\boldsymbol{c}_3, \boldsymbol{c}_1)$ | $(\boldsymbol{c}_3, \boldsymbol{c}_1)$ | $(\boldsymbol{0}, \boldsymbol{0})$ | $\sqrt{q}$ |
| $V_4$ | $(\boldsymbol{0}, -2\boldsymbol{c}_1)$ | $(\boldsymbol{c}_2, \boldsymbol{c}_1)$ | $(\boldsymbol{c}_2, \boldsymbol{c}_1)$ | $(\boldsymbol{0}, \boldsymbol{0})$ | $(\boldsymbol{0}, \boldsymbol{0})$ | $(\boldsymbol{0}, \boldsymbol{0})$ | $q^{1/4}$ |
| $V_5$ | $(\boldsymbol{0}, -2\boldsymbol{c}_1)$ | $(\boldsymbol{0}, \boldsymbol{0})$ | $(\boldsymbol{0}, \boldsymbol{0})$ | $(\boldsymbol{0}, \boldsymbol{0})$ | $(\boldsymbol{0}, \boldsymbol{0})$ | $(\boldsymbol{0}, -2\boldsymbol{c}_1)$ | $1$ |

Note that the $H_3$ and $V_3$ tiles are characterized by three color labels. This is because the solid (resp. dashed) arrows in $H_3$ (resp. $V_3$) in general can be of different color, since they originate from different spins. The dashed (resp. solid) arrows in $H_3$ (resp. $V_3$) are always of the same color, as they originate from the same spins.

All of the tiles giving nonzero entries of the tensors satisfy

$$\sum_{i \in \text{I}} \boldsymbol{k}_i = \sum_{o \in \text{O}} \boldsymbol{k}_o, \tag{3.2}$$

where I and O refers to the set of indices that are incoming (I) or outgoing (O), which differs for solid and dashed arrows, according to panel (c) of Fig. (9). The final step is then to weigh the tiles correctly in terms of the deformation parameter $q$. The correct weights are derived in Appendix C, using the U(1) invariance of the tensors, but basically amounts to counting up the various tiles in the different valid tilings and distributing factors of $q$ so that the weighing of states in Eq. (2.5) is achieved. Since each height vertex in the 6-vertex model has four adjacent spins, compared to just two adjacent spins in the single Fredkin chain, the weight of the $X_i$ tile is equal to the square root of the weight of the corresponding $A_i$ tile in the single Fredkin case. This gives the following expression for the tensors

$$X(q) = \sum_{\boldsymbol{c}_{1,2,3} = \{(1,0),\,(0,1)\}} \left[ X_1(\boldsymbol{c}_1) + q^{\frac{1}{4}} X_2(\boldsymbol{c}_1, \boldsymbol{c}_2) + \sqrt{q} X_3(\boldsymbol{c}_1, \boldsymbol{c}_2, \boldsymbol{c}_3) + q^{\frac{1}{4}} X_4(\boldsymbol{c}_1, \boldsymbol{c}_2) + X_5(\boldsymbol{c}_1) \right],$$
$$\tag{3.3}$$

where $X = H, V$ and we consider the case of two colors, red $\boldsymbol{c}_i = (1,0)$ and blue $\boldsymbol{c}_i = (0,1)$. The TN representation of the GS (2.5) is given in Fig. (10) for the $L = 4$ system. The contraction between the $H(q)$-tensors and $V(q)$-tensors on the same level $l$ is through the pairs $\boldsymbol{k}_3 - \boldsymbol{k}_5$, $\boldsymbol{k}_2 - \boldsymbol{k}_4$. In the figure, contractions over these pairs of indices are indicated by squares sharing edges. Contractions between $H(q)$-tensors ($V(q)$-tensors) on different levels is through the pair $\boldsymbol{k}_1 - \boldsymbol{k}_6$, and are indicated by a small black dot at the center of

the squares at the $l = 1$ level. Note that the boundary tensor $\delta_{\boldsymbol{k}_i,(0,0,0,0)}$ ensures that no arrows flow out of the TN, while the boundary tensor $\delta_{\boldsymbol{k}_i,(\boldsymbol{c}_1,\boldsymbol{c}_2)}$ at the top level, fixes pairs of arrows to flow into and out from the outer walls of the tensors, at the even numbered spin chains. These are seen as the black arrows in Fig. (7). The color of these arrows are arbitrary, and one can either sum over possible color configurations or fix them to a given color.

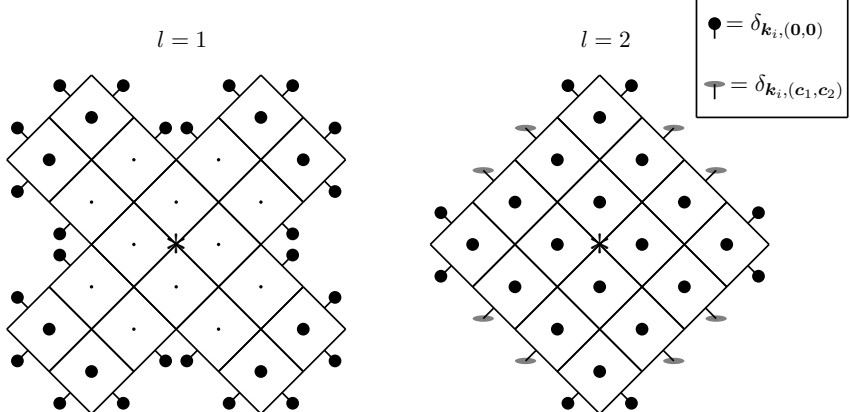

Figure 10: TN representation of the GS (2.5) of an $L = 4$ system. The levels $l = 1$ (bottom level) and $l = 2$ (top level) are depicted separately, and are contracted through $\boldsymbol{k}_6$ indices at level $l = 1$ and $\boldsymbol{k}_1$ indices at level $l = 2$, as indicated by the small black dots. The asterisk $*$ is a visual aid marking the center of the levels, and the $l = 2$ level should be placed on top of the $l = 1$ level so that the asterisks are matching. The box in the top right corner shows the two different types of ancilla tensors, where $\boldsymbol{c}_1$ and $\boldsymbol{c}_2$ are the arbitrary colors of the spins outside our system. The large black dots in the center of the squares at $l = 2$ indicate that the $\boldsymbol{k}_6$ index is contracted with $\delta_{\boldsymbol{k}_6,(\boldsymbol{0},\boldsymbol{0})}$. The physical legs of the TN points into the paper plane.

# 4 TN for polychromatic lozenge tiling

## 4.1 Prismatic tiles

We now obtain the TN representation of the GS (2.8) of the lozenge tiling coupled Fredkin chains, by using the tiling method. We associate a tower of tiles for each triangle in the lattice, see Fig. (11) for a top view. The intuition behind the choice is that in the monochromatic case, the TN would only need to guarantee the tiling of lozenges, or dimer-covering of a hexagonal lattice. This can then be represented by a PEPS, just like how the colorless 6-vertex ground state, can be represented by a PEPS with quadrivalent tensors [36]. When the physical degrees of freedom is enlarged with colors, instead of planar square tensors in the PEPS, we considered cube tensors in the previous section, enabling the connection of entangled pairs farther away more easily in the TN. Therefore, following the same reasoning, we add depth to triangles making them into prisms, which will be the tensor of choice in this section. Three representative tiles corresponding to three of the 9 nonzero entries of a tensor is given in Fig. (12). Since the hexagonal lattice is bipartite, there are two different types of tensors, which project to right- and left-pointing triangles, denoted $R_i$ and $L_i$ respectively as shown in Fig. (11). In a 2D lozenge tiling, a lozenge covers two triangles, one of each type. A single lozenge also carries two up-down

degrees of freedom, belonging to Fredkin paths in two different planes. This means the prisms carry the information of one up-down move per triangle in the 2D lattice, compared to the cubes in the previous section that corresponded to one spin in Fig. (4). We will be referring to the triangles, or half lozenges, as "spins" for the rest of the section.

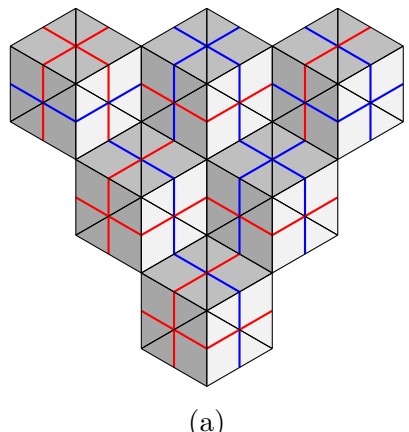
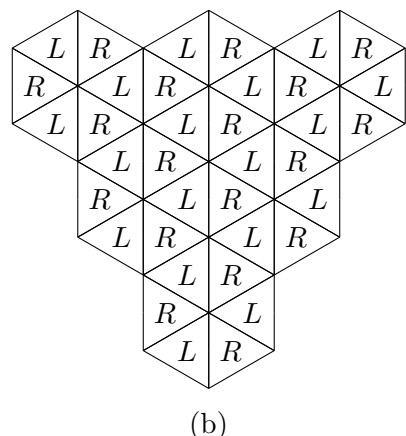

| (a) | (b) |

Figure 11: (a) Top view of tower of prism tiles associated to each triangle in the lattice, giving two tower of prism tiles per lozenge. (b) The two sublattices corresponding to right-pointing prism tiles $R$ and left-pointing prism tiles $L$.

The physical configurations in the GS of the lozenge tiling model are described by color correlated Dyck walks, just as the physical configurations where in the GS of the single and 6-vertex coupled Fredkin chains. This motivates a set of tiles similar to the set of square and cubical tiles for the single and 6-vertex coupled Fredkin chains respectively. In the tiles for the Fredkin model, only one type of arrow was used while in the 6-vertex model we used two types of arrows due to Dyck walks in two types of spin chains (horizontal and vertical). In the GS of the quantum lozenge model, physical configurations are characterized by three different types of Dyck walks, which calls for the introduction of a third type of arrow, shown with dotted lines. We use solid arrows for Dyck walks in the $xy$-plane, dashed arrows for Dyck walks in the $xz$-plane and dotted arrows for Dyck walks in the $yz$-plane. The tiles can be seen in Fig. (12).

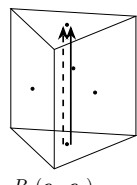
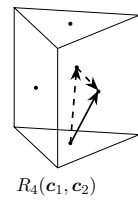
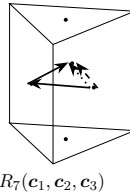
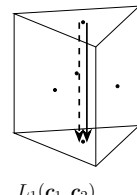
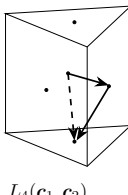
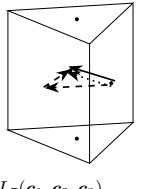

$R_1(\boldsymbol{c_1}, \boldsymbol{c_2})$    $R_4(\boldsymbol{c_1}, \boldsymbol{c_2})$    $R_7(\boldsymbol{c_1}, \boldsymbol{c_2}, \boldsymbol{c_3})$    $L_1(\boldsymbol{c_1}, \boldsymbol{c_2})$    $L_4(\boldsymbol{c_1}, \boldsymbol{c_2})$    $L_7(\boldsymbol{c_1}, \boldsymbol{c_2}, \boldsymbol{c_3})$

Figure 12: Three representatives of both right pointing prism tiles $R_i$ and left-pointing prism tiles $L_i$ corresponding to nonzero entries of a tensor in the TN representation of the GS of the lozenge tiling model. The color labels $c_i$ denote the color of the arrows and arrows of the same type within a prism tile always have the same color. The black dots mark the center of each face of the prism tile. A top view of the complete set of prism tiles are seen in Fig. (13).

As in the 6-vertex case, we associate two arrows per spin, now necessary for ensuring that there is only one tiling of the prisms per lozenge tiling. But unlike the 6-vertex case, we associate two different arrows which in general point in opposite directions as well as in the same direction. This is motivated by ensuring that prisms always pair up in such

a way that they correspond to one of the three lozenges, as shown in Appendix D.1. This difference between the cubical tiles and the prism tiles, can be understood by the fact that each lozenge participates in two different Dyck walks, while the spins in the 6-vertex model only participates in one Dyck walk. The two prism tiles corresponding to a lozenge should therefore indicate the entanglement in the directions of both the Dyck walks.

From now on, the prisms will be presented as triangles, their projection onto the physical plane. One working choice of legal tiles are listed in Fig. (13), where ⊗ and ⊛ (resp. ⊙ and ⊙) means only one arrow of that type pointing into (resp. out from) from the paper plane, as opposed to meaning two arrows of that type in Fig. (6). In addition we have the symbols ⋅⋅ and ⋰ meaning a dotted arrow pointing out of and into the paper plane respectively. It goes without saying that there are other equivalent choices of the set of valid tiles, related to our choice by a gauge transformation, such as reversing all the arrows of a certain type.

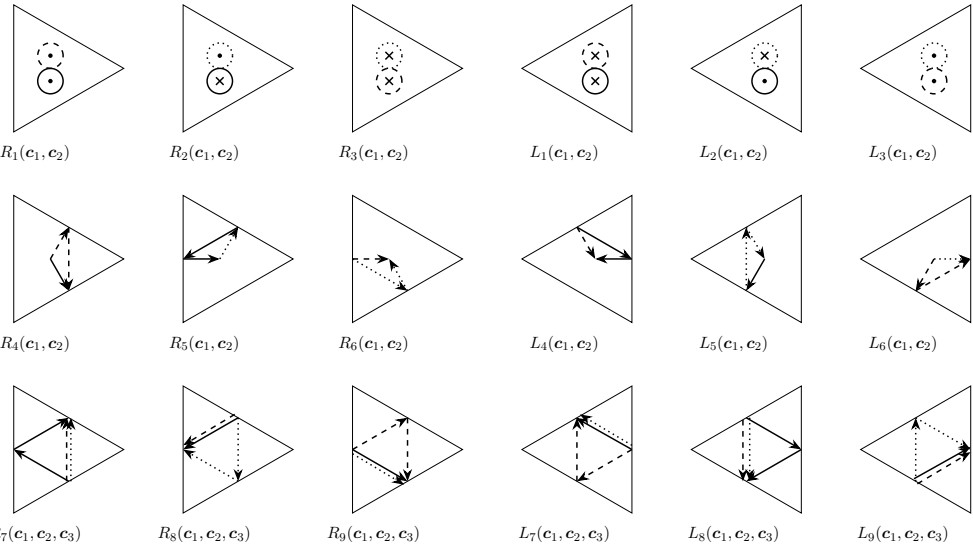

Figure 13: Top view of the set of prism tiles. Note that arrows of the same type within a tile always have the same color. The full prism drawing of the tiles $Y_1(c_1, c_2)$, $Y_4(c_1, c_2)$ and $Y_7(c_1, c_2, c_3)$, with $Y = R, L$ is seen in Fig. (12).

## 4.2 Tilings of prisms

We now illustrate the one-to-one correspondence between valid 3D tilings of prisms and the 2D tilings of lozenges $|T^C\rangle$ in the GS (2.8). As an example, we show the corresponding 3D tilings topographically for the minimal and maximal height lozenge tilings for a small system in Fig. (14). Again, the full 3D tilings are understood as the placement of the $l = 2$ level on top of the $l = 1$ levels so that the asterisk $*$ are matching. Note that we use a third color, green, to better illustrate the correlations.

As in the previous section, the bijection is established by noticing that in each tower of prisms, there is a certain layer depending on the physical 2D configuration, on which the prism has three arrows, namely those of type $Y_{4,5,6}$. Above that level, all the tiles have 4 horizontal arrows, and below they all have two vertical ones. The argument, using the height function $\phi$, is presented in Appendix D.

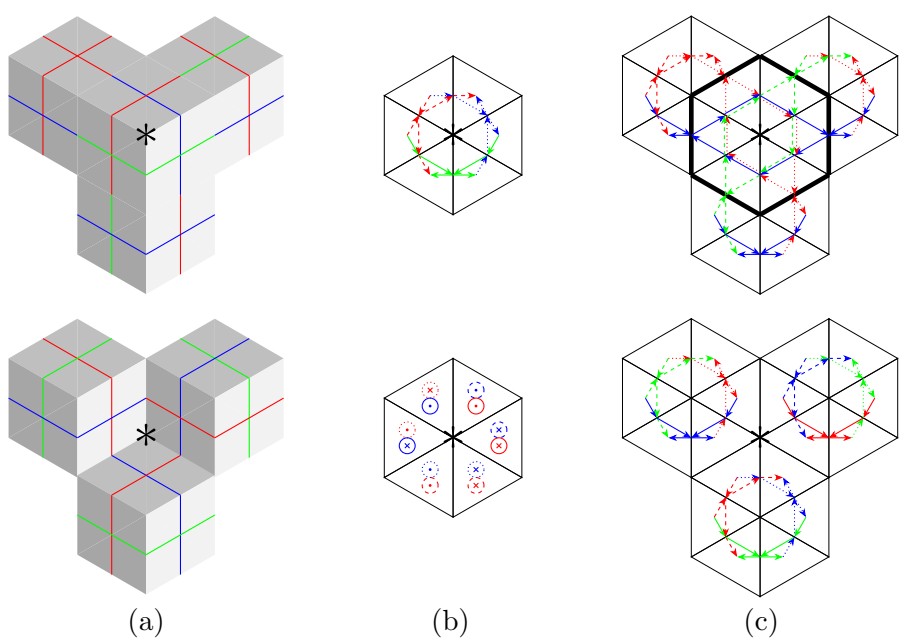

(a)          (b)          (c)

Figure 14: The highest (upper panels) and lowest (lower panels) height configurations with a certain coloring for a small system size: (a) The lozenge tilings. (b) The prism tilings in the $l = 1$ level. (c) The prism tilings in the $l = 2$ level. The asterisk $*$ is a visual aid marking the center. The hexagon marked with thick lines is filled with tiles with 4 arrows, and is discussed in Appendix D.2. We have used a third color, green, to better indicate color correlations.

## 4.3 From prismatic tiles to 5-leg tensors

We denote tensors populating the right pointing (resp. left-pointing) sublattice with $R(q)$ (resp. $L(q)$). Their indices are labeled $\boldsymbol{k}_1, \ldots, \boldsymbol{k}_5$ as specified in Fig. (15). This choice of indexing, implies that adjacent $R(q)$ and $L(q)$ tensors are contracted through indices $\boldsymbol{k}_i$ of the same subscript for $i = 2, 3, 4$, and that the indices $\boldsymbol{k}_1$ and $\boldsymbol{k}_5$ are contracted with each other between adjacent tensors of the same type in a tower.

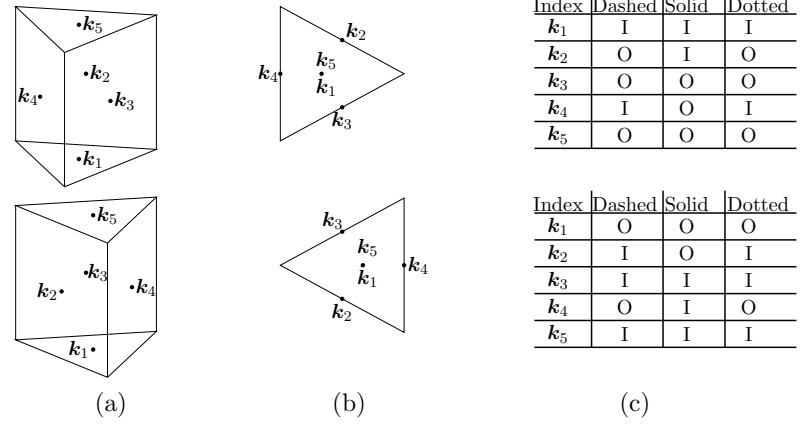

| Index | Dashed | Solid | Dotted |
|---|---|---|---|
| $\boldsymbol{k}_1$ | I | I | I |
| $\boldsymbol{k}_2$ | O | I | O |
| $\boldsymbol{k}_3$ | O | O | O |
| $\boldsymbol{k}_4$ | I | O | I |
| $\boldsymbol{k}_5$ | O | O | O |

| Index | Dashed | Solid | Dotted |
|---|---|---|---|
| $\boldsymbol{k}_1$ | O | O | O |
| $\boldsymbol{k}_2$ | I | O | I |
| $\boldsymbol{k}_3$ | I | I | I |
| $\boldsymbol{k}_4$ | O | I | O |
| $\boldsymbol{k}_5$ | I | I | I |

(a)          (b)          (c)

Figure 15: (a) - (b) Depiction of the indices $\boldsymbol{k}_1, \ldots, \boldsymbol{k}_5$ and the corresponding prism faces, where (b) indicates this in the top view of the prism. (c) Classification of the the indices in terms of incoming (I) and outgoing (O) with respect to each arrow type. The top (resp. bottom) row depicts the $R_i$ (resp. $L_i$) prisms.

As in the previous section, the tiles here are still source- and drain-free, for each type of arrows individually. But unlike the previous section, the roles of incoming and outgoing arrows are not fixed for a certain type of arrow in a given type of tile. Due to the intrinsic geometric frustration of the triangular lattice, there is not a unique choice of labeling the indices as incoming or outgoing that is more natural or symmetric than the others. This leads to the indices taking on a larger range of possible values. We choose a working definition in panel (c) in Fig. (15), which is opposite for the $R(q)$ and $L(q)$ tensors, such that outgoing indices are contracted with incoming and vice versa. Extending the notation for the 6-vertex case, we specify the indices with a 6-component vector, where the last two components encode the colors of the dotted arrow, that is $\boldsymbol{k}_i = (\boldsymbol{c}_{\text{dashed}}, \boldsymbol{c}_{\text{solid}}, \boldsymbol{c}_{\text{dotted}})$. As before, we have $\boldsymbol{c}_{\text{type}} = (1, 0)$ or $\boldsymbol{c}_{\text{type}} = (0, 1)$ for a single red and blue arrow of the given type and $\boldsymbol{c}_{\text{type}} = (\boldsymbol{0}) = (0, 0)$, corresponding to no arrows of the given type. The major difference from the 6-vertex case is that $\boldsymbol{k}_2$, $\boldsymbol{k}_3$ or $\boldsymbol{k}_4$, can now have two arrows of the same type and color, but flowing in opposite directions. This for instance corresponds to the dashed arrows at the $\boldsymbol{k}_2$-index of the $R_4$ tile in Fig. (12) and Fig. (13). We label the corresponding index value with $\boldsymbol{0^c}$ for a given color $\boldsymbol{c} = (1, 0)$ or $(0, 1)$ of the two arrows. Eq. (3.2) is recovered if we consider $\boldsymbol{0^c}$ as a zero. This can be interpreted as a degeneracy in the index corresponding to the particle number $n = 0$ [35]. This approach gives the index values seen in Table. 2 for the various tiles.

Table 2: Tables displaying the values of the index configurations giving a non-zero value of the rank-5 tensors $Y(q) = R(q), L(q)$ and the corresponding tile $Y_i$.

| Tile | $\boldsymbol{k}_1$ | $\boldsymbol{k}_2$ | $\boldsymbol{k}_3$ | $\boldsymbol{k}_4$ | $\boldsymbol{k}_5$ | Value |
|------|------|------|------|------|------|-------|
| $Y_1$ | $(\boldsymbol{c}_1, \boldsymbol{c}_2, \boldsymbol{0})$ | $(\boldsymbol{0}, \boldsymbol{0}, \boldsymbol{0})$ | $(\boldsymbol{0}, \boldsymbol{0}, \boldsymbol{0})$ | $(\boldsymbol{0}, \boldsymbol{0}, \boldsymbol{0})$ | $(\boldsymbol{c}_1, \boldsymbol{c}_2, \boldsymbol{0})$ | 1 |
| $Y_2$ | $(\boldsymbol{0}, -\boldsymbol{c}_1, \boldsymbol{c}_2)$ | $(\boldsymbol{0}, \boldsymbol{0}, \boldsymbol{0})$ | $(\boldsymbol{0}, \boldsymbol{0}, \boldsymbol{0})$ | $(\boldsymbol{0}, \boldsymbol{0}, \boldsymbol{0})$ | $(\boldsymbol{0}, -\boldsymbol{c}_1, \boldsymbol{c}_2)$ | 1 |
| $Y_3$ | $(-\boldsymbol{c}_1, \boldsymbol{0}, -\boldsymbol{c}_2)$ | $(\boldsymbol{0}, \boldsymbol{0}, \boldsymbol{0})$ | $(\boldsymbol{0}, \boldsymbol{0}, \boldsymbol{0})$ | $(\boldsymbol{0}, \boldsymbol{0}, \boldsymbol{0})$ | $(-\boldsymbol{c}_1, \boldsymbol{0}, -\boldsymbol{c}_2)$ | 1 |

| Tile | $\boldsymbol{k}_1$ | $\boldsymbol{k}_2$ | $\boldsymbol{k}_3$ | $\boldsymbol{k}_4$ | $\boldsymbol{k}_5$ | Value |
|------|------|------|------|------|------|-------|
| $R_4$ | $(\boldsymbol{c}_1, \boldsymbol{c}_2, \boldsymbol{0})$ | $(\boldsymbol{0^{c_1}}, \boldsymbol{0}, \boldsymbol{0})$ | $(\boldsymbol{c}_1, \boldsymbol{c}_2, \boldsymbol{0})$ | $(\boldsymbol{0}, \boldsymbol{0}, \boldsymbol{0})$ | $(\boldsymbol{0}, \boldsymbol{0}, \boldsymbol{0})$ | $q^{\frac{1}{4}}$ |
| $R_5$ | $(\boldsymbol{0}, -\boldsymbol{c}_1, \boldsymbol{c}_2)$ | $(\boldsymbol{0}, \boldsymbol{c}_1, \boldsymbol{c}_2)$ | $(\boldsymbol{0}, \boldsymbol{0}, \boldsymbol{0})$ | $(\boldsymbol{0}, \boldsymbol{0^{c_1}}, \boldsymbol{0})$ | $(\boldsymbol{0}, \boldsymbol{0}, \boldsymbol{0})$ | $q^{\frac{1}{4}}$ |
| $R_6$ | $(-\boldsymbol{c}_1, \boldsymbol{0}, -\boldsymbol{c}_2)$ | $(\boldsymbol{0}, \boldsymbol{0}, \boldsymbol{0})$ | $(\boldsymbol{0}, \boldsymbol{0}, \boldsymbol{0^{c_2}})$ | $(\boldsymbol{c}_1, \boldsymbol{0}, \boldsymbol{c}_2)$ | $(\boldsymbol{0}, \boldsymbol{0}, \boldsymbol{0})$ | $q^{\frac{1}{4}}$ |
| $R_7$ | $(\boldsymbol{0}, \boldsymbol{0}, \boldsymbol{0})$ | $(\boldsymbol{c}_1, -\boldsymbol{c}_2, \boldsymbol{c}_3)$ | $(-\boldsymbol{c}_1, -\boldsymbol{c}_2, -\boldsymbol{c}_3)$ | $(\boldsymbol{0}, \boldsymbol{0^{c_2}}, \boldsymbol{0})$ | $(\boldsymbol{0}, \boldsymbol{0}, \boldsymbol{0})$ | $\sqrt{q}$ |
| $R_8$ | $(\boldsymbol{0}, \boldsymbol{0}, \boldsymbol{0})$ | $(-\boldsymbol{c}_1, \boldsymbol{c}_2, -\boldsymbol{c}_3)$ | $(\boldsymbol{0}, \boldsymbol{0}, \boldsymbol{0^{c_3}})$ | $(-\boldsymbol{c}_1, \boldsymbol{c}_2, -\boldsymbol{c}_3)$ | $(\boldsymbol{0}, \boldsymbol{0}, \boldsymbol{0})$ | $\sqrt{q}$ |
| $R_9$ | $(\boldsymbol{0}, \boldsymbol{0}, \boldsymbol{0})$ | $(\boldsymbol{0^{c_1}}, \boldsymbol{0}, \boldsymbol{0})$ | $(\boldsymbol{c}_1, \boldsymbol{c}_2, \boldsymbol{c}_3)$ | $(\boldsymbol{c}_1, -\boldsymbol{c}_2, \boldsymbol{c}_3)$ | $(\boldsymbol{0}, \boldsymbol{0}, \boldsymbol{0})$ | $\sqrt{q}$ |
| $L_4$ | $(\boldsymbol{c}_1, \boldsymbol{c}_2, \boldsymbol{0})$ | $(\boldsymbol{0}, \boldsymbol{0}, \boldsymbol{0})$ | $(\boldsymbol{c}_1, \boldsymbol{c}_2, \boldsymbol{0})$ | $(\boldsymbol{0}, \boldsymbol{0^{c_2}}, \boldsymbol{0})$ | $(\boldsymbol{0}, \boldsymbol{0}, \boldsymbol{0})$ | $q^{\frac{1}{4}}$ |
| $L_5$ | $(\boldsymbol{0}, -\boldsymbol{c}_1, \boldsymbol{c}_2)$ | $(\boldsymbol{0}, \boldsymbol{c}_1, \boldsymbol{c}_2)$ | $(\boldsymbol{0}, \boldsymbol{0}, \boldsymbol{0^{c_2}})$ | $(\boldsymbol{0}, \boldsymbol{0}, \boldsymbol{0})$ | $(\boldsymbol{0}, \boldsymbol{0}, \boldsymbol{0})$ | $q^{\frac{1}{4}}$ |
| $L_6$ | $(-\boldsymbol{c}_1, \boldsymbol{0}, -\boldsymbol{c}_2)$ | $(\boldsymbol{0^{c_1}}, \boldsymbol{0}, \boldsymbol{0})$ | $(\boldsymbol{0}, \boldsymbol{0}, \boldsymbol{0})$ | $(\boldsymbol{c}_1, \boldsymbol{0}, \boldsymbol{c}_2)$ | $(\boldsymbol{0}, \boldsymbol{0}, \boldsymbol{0})$ | $q^{\frac{1}{4}}$ |
| $L_7$ | $(\boldsymbol{0}, \boldsymbol{0}, \boldsymbol{0})$ | $(\boldsymbol{0^{c_1}}, \boldsymbol{0}, \boldsymbol{0})$ | $(-\boldsymbol{c}_1, -\boldsymbol{c}_2, -\boldsymbol{c}_3)$ | $(-\boldsymbol{c}_1, \boldsymbol{c}_2, -\boldsymbol{c}_3)$ | $(\boldsymbol{0}, \boldsymbol{0}, \boldsymbol{0})$ | $\sqrt{q}$ |
| $L_8$ | $(\boldsymbol{0}, \boldsymbol{0}, \boldsymbol{0})$ | $(-\boldsymbol{c}_1, \boldsymbol{c}_2, -\boldsymbol{c}_3)$ | $(\boldsymbol{c}_1, \boldsymbol{c}_2, \boldsymbol{c}_3)$ | $(\boldsymbol{0}, \boldsymbol{0^{c_2}}, \boldsymbol{0})$ | $(\boldsymbol{0}, \boldsymbol{0}, \boldsymbol{0})$ | $\sqrt{q}$ |
| $L_9$ | $(\boldsymbol{0}, \boldsymbol{0}, \boldsymbol{0})$ | $(\boldsymbol{c}_1, -\boldsymbol{c}_2, \boldsymbol{c}_3)$ | $(\boldsymbol{0}, \boldsymbol{0}, \boldsymbol{0^{c_3}})$ | $(\boldsymbol{c}_1, -\boldsymbol{c}_2, \boldsymbol{c}_3)$ | $(\boldsymbol{0}, \boldsymbol{0}, \boldsymbol{0})$ | $\sqrt{q}$ |

By weighing the different tiles with an appropriate factor of $q$ we can finally arrive at the TN representation. The weights can again be determined by computing the volume $V$ in Eq. (2.6) of lozenge tilings, then counting the number of various tiles in the corresponding valid tiling and then distributing the factors of $q$ to the various tiles. This is done in Appendix E, where it is found that in order to get the correct weighing of lozenge tilings using the tile weights in Table. 2, we must weigh some of the boundary tensors with a factor of $q$. The TN then becomes as seen in Fig. (16) for a small system where the tensors

$R(q)$ and $L(q)$ is given by

$$Y(q) = \sum_{i=1}^{3} \sum_{\boldsymbol{c}_1, \boldsymbol{c}_2} Y_i(\boldsymbol{c}_1, \boldsymbol{c}_2) + \sum_{i=4}^{6} \sum_{\boldsymbol{c}_1, \boldsymbol{c}_2} q^{1/4} Y_i(\boldsymbol{c}_1, \boldsymbol{c}_2) + \sum_{i=7}^{9} \sum_{\boldsymbol{c}_1, \boldsymbol{c}_2, \boldsymbol{c}_3} \sqrt{q} Y_i(\boldsymbol{c}_1, \boldsymbol{c}_2, \boldsymbol{c}_3), \quad (4.1)$$

where $Y(q) = R(q), L(q)$. The boundary tensors seen in Fig. (16) ensure that no arrows flow out.

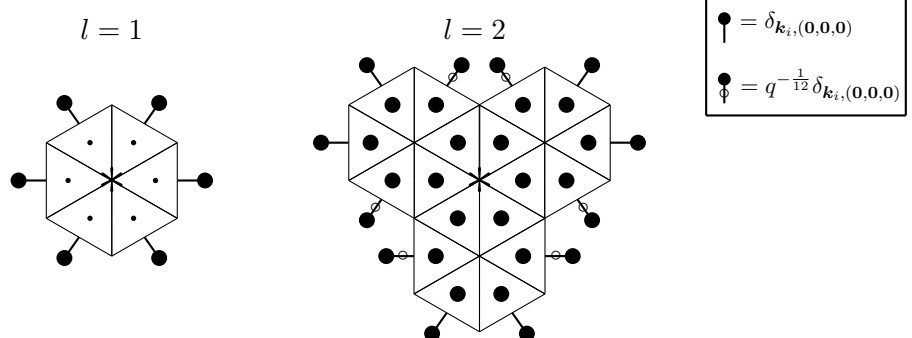

Figure 16: TN representation of the GS (2.8), for a small system size. The levels $l = 1$ (bottom level) and level $l = 2$ (top level) are depicted separately, and are contracted through $\boldsymbol{k}_5$ indices at level $l = 1$ and $\boldsymbol{k}_1$ indices at level $l = 2$, indicated by small black dots. The asterisk $*$ is a visual aid marking the center of the levels, and the $l = 2$ level should be placed on top of the $l = 1$ level so that the asterisks are matching. The box in the top right shows the ancilla tensors, where one is weighed with $q^{-\frac{1}{12}}$, see Appendix E. The large black dots in the center of the squares at $l = 2$ indicate that the $\boldsymbol{k}_5$ index is contracted with the unweighted ancilla tensor. The physical legs of the TN points into the paper plane.

## 5 Correlation functions

The holographic TN descriptions of the highly entangled GSs given in the previous sections merely reflect upper bounds on the scaling of EE. In fact, all of the GSs go through a quantum phase transition as the deformation parameter $q$ is tuned across the critical value $q = 1$. In the polychromatic case, the phase transition is first and foremost characterized by the EE and spectral gap scaling. As will be shown in Sec. 5.2, the phase transition also manifests itself in the color-correlation function. But even when the color degree of freedom is not in play, the phase transition is a typical order-disorder phase transition and can be characterized by a spin order parameter.

### 5.1 Spin correlation

Order parameters in translationally invariant systems without boundaries are usually not spatially varying. In our GSs, the boundary spins are fixed in opposite directions on opposite sides of the system. The order parameter therefore becomes a spatially varying field that decays from the boundary value within a length scale called the "skin depth" that reflects the correlation length. The correlation length is typically obtained from computing the two-point correlation functions, which has been done in 1D for the critical point $q = 1$ [38, 39]. However, for the other phases and for 2D in general, exact evaluation

becomes inaccessible, but we can probe the correlation from the boundary effect on the order parameter. The magnetization as a function of the spatial coordinate is simply the derivative of the height function

$$\langle S_r \rangle \propto \partial_r \langle \phi_r \rangle, \tag{5.1}$$

where $r$ is the distance from the spin to the closest boundary spin (belonging to the same sublattice in the 2D case). The scaling behavior of the height function has been heavily studied with a variety of methods [12, 13, 15–17, 40]. Since the generalization to 2D, it is now well understood that the scaling limit of such models in any dimension are described by random surfaces subject to a hard wall constraint from below [19, 41–43]. These results are summarized in Table. 3, where the only new result is the decay behavior of the $q < 1$ phase from the boundary value to the previously known bulk value 0, which we now argue. First, although a ground state that obeys the area law of EE not necessarily imply a finite spectral gap [1], the $q < 1$ phase of the 1D monochromatic Motzkin chain was recently shown to be gapped [44]. We expect the same to hold for our 2D generalizations even with a color degree of freedom. But even without knowledge of how adding color changes the gap, we could still conclude from a finite gap in the spin sector that the spin-spin correlation decays exponentially. Second, one can consider the rate of change in the expectation value of height $\partial_r \langle \phi_r \rangle$ at a certain distance $r$ away from the boundary, using the law of total probability. Denoting the probability distribution of the height function at distant $r$ as $p_r(\phi_r)$, and the probability of having a step upward right after as $p^+(\phi_r)$ which depends on the current height, we can write

$$
\begin{aligned}
\langle \phi_{r+dr} \rangle &= \int_0^{r+dr} d\phi_{r+dr} \, p_{r+dr}(\phi_{r+dr}) \phi_{r+dr} \\
&= \int_0^r d\phi_r \int_{\max\{\phi_r - dr, 0\}}^{\min\{\phi_r + dr, r\}} d\phi_{r+dr} \, p(\phi_{r+dr} | \phi_r) p_r(\phi_r) \phi_{r+dr} \\
&= \int_0^r d\phi_r \int_{\max\{-dr, -\phi_r\}}^{\min\{dr, r - \phi_r\}} d\delta\phi_r \, p(\delta\phi_r) p_r(\phi_r)(\phi_r + \delta\phi_r) \\
&= \int_0^r d\phi_r \, p_r(\phi_r)\phi_r + \int_0^r d\phi_r \, p(\phi_r) \int_{\max\{-dr, -\phi_r\}}^{\min\{dr, r - \phi_r\}} d\delta\phi_r \, p(\delta\phi_r)\delta\phi_r \\
&= \langle \phi_r \rangle + \langle \overline{\delta\phi_r} \rangle,
\end{aligned}
$$

where we have used brackets to denote averaging over the random surfaces in the GS, and the overline to denote the average incremental change. This simply says that the rate of change in the expectation value of height is proportional to the probability bias $\partial_r \langle \phi_r \rangle = \langle \partial_r \overline{\delta\phi_r} \rangle$. The latter is a negative value that depends on the current height $\phi_r$, which has larger magnitude if $\phi_r$ is larger. This is because the hard wall constraint from below compromises the effect of the deformation that favors lower height. When the current height is large enough that a down step is way clear of hitting the wall at 0 height, the existence of the hard wall can be neglected. And its effect gets more and more prominent as the height gets closer to 0. To extract the long-distance behavior of the height function deep in the bulk, we can Taylor expand the bias as $\langle \partial_r \overline{\delta\phi_r} \rangle = A - B\langle \phi_r \rangle + C\langle \phi_r^2 \rangle + \cdots$, with the constant $B > 0$. Now, since on average the height stays close to 0 in the $q < 1$ phase, we know in the scaling limit $A = 0$, and since $\phi_r \ll 1$, we can drop the higher order terms to the first order of approximation. Then the differential equation is solved by $\langle \phi_r \rangle \sim e^{-Br}$. This completes the picture of the height scaling, which is seen in the left panel of Table. 3 for all $q$.

---

[1]Among plenty of others, one counterexample is the frustration-free 6-vertex model without the color degree of freedom, where the vanishing spectral gap was rigorously proven in Ref. [36].

Table 3: Scaling of the height function $\phi_r$ (left panel) and of the spatially varying spin order parameter $\langle S_r \rangle$ (right panel) in 1D and 2D for different values of the deformation parameter $q$.

| | 1D | 2D | $q$ |
|---|---|---|---|
| | $r$ | $r$ | $q > 1$ |
| $\phi_r \sim$ | $\sqrt{r}$ | $\log r$ | $q = 1$ |
| | $e^{-\alpha r}$ | $e^{-\beta r}$ | $q < 1$ |

| | 1D | 2D | $q$ |
|---|---|---|---|
| | const. | const. | $q > 1$ |
| $\langle S_r \rangle \sim$ | $r^{-\frac{1}{2}}$ | $r^{-1}$ | $q = 1$ |
| | $e^{-\alpha r}$ | $e^{-\beta r}$ | $q < 1$ |

From the height scaling in Table. 3, we can get the spatially varying spin order parameter $\langle S_r \rangle$ through Eq. (5.1). This is seen in the right panel of Table. 3. Notice the similarity with the usual scaling behavior of two-point correlation functions in both the ordered and disorder phase and at the critical point. This should be compared with the scaling of the spin-gap $\Delta_s(L)$, i.e. the scaling of the energy needed to create an excitation in the spin sector for a system of linear size $L$, seen in Table. 4.

Table 4: Scaling of the spectral gap in the spin sector $\Delta_s(L)$ in 1D and 2D for different values of the deformation parameter $q$.

| | 1D | 2D | $q$ |
|---|---|---|---|
| | $< e^{-C_1 L}$ [17, 45] | $< e^{-C_2 L^2}$ [18] | $q > 1$ |
| $\Delta_s(L)$ | $\sim L^{-C}$ [31] | **Conjecture:** $\sim L^{-C'}$ | $q = 1$ |
| | $\sim$ const. [44] | **Conjecture:** $\sim$ const. | $q < 1$ |

## 5.2 Color correlation

Unlike the spin sector, there is no fixed boundary condition on the color degrees of freedom. So to reveal a phase transition, we must evaluate the color correlation function. This is relatively easy to do for the correlation with the boundary spin. We introduce a color operator $\hat{c}$ with eigenvalues $\pm 1$ for red and blue spins respectively. The Hamiltonian is symmetric in the color degree of freedom and all spin configurations in the GS are described by random surfaces where the height remain non-negative. Furthermore, the spin configurations in the GS are described by color matching between nearest up-down spin pairs at the same height. This gives

$$G_c(r) = \langle \hat{c}_0 \hat{c}_r \rangle = p(c_0, c_r \text{ forms a nearest up-down pair}). \quad (5.2)$$

In the $q < 1$ phase, the volume weighted random surfaces stay close to height 0 with large probability, meaning that touching the hard wall at the bottom is an event that happens at each step independently at a constant rate $p^\circ$. The distance $r^*$ from the boundary at which point the height reaches 0 for the first time therefore obeys an exponential distribution. Indeed, the probability of $c_r$ being the nearest spin that forms a pair with $c_0$ at height 0 is thus $(1 - p^\circ)^{r-1} p^\circ$, which makes the color correlation decay exponentially. Here, it is important for the scaling that $p^\circ$ is not a small quantity, otherwise the powers of $1 - p^\circ$ would be close to 1 and not render exponentially decaying factor.

At the critical point $q = 1$, we can exactly enumerate the Dyck paths to evaluate the probability of the shortest distance from the boundary where the height reaches 0 being

$r^*$ for the 1D models. Since the number of Dyck paths of distance $r^*$ is

$$N_{\text{Dyck}}(r^*) = \frac{2}{r^*+2}\binom{r^*}{\frac{r^*}{2}},$$ (5.3)

the probability that gives the color correlation is given by

$$p(c_0, c_r^* \text{ forms a nearest up-down pair}) = \frac{N_{\text{Dyck}}(r^*)N_{\text{Dyck}}(L-r^*)}{N_{\text{Dyck}}(L)}.$$ (5.4)

Using Sterling's approximation, in the $L, r \gg 1$ limit, this becomes

$$\langle \hat{c}_0 \hat{c}_r \rangle \sim \frac{r^{-\frac{3}{2}}(L-r)^{-\frac{3}{2}}}{L^{-\frac{3}{2}}},$$ (5.5)

which gives the power law decay $\langle \hat{c}_0 \hat{c}_r \rangle \sim r^{-\frac{3}{2}}$ when the $L \gg r$ limit is taken.

In 2D, such an enumeration of random surfaces is not accessible, but we can nevertheless argue the power law decay of color correlation in the scaling limit. The height distribution at any location in the bulk obeys Gaussian distribution [33, 46], with or without the hard wall constraint. The variance scales with the distance to the boundary as $\log r$, which can be easily computed by Fourier transforming the Gaussian propagator. The average height, on the other hand, scales as $\log r$, due to the entropic repulsion of the hard wall [42]. The probability of hitting height 0, a deviation $\sim \log r$ away from the average, regardless of whether it is the first time or not, scales as

$$p_0(r) \sim e^{-g\frac{(0-\log r)^2}{\log r}} = r^{-\eta},$$ (5.6)

where $g$ is a constant that combines the exact coefficients from both the average and variance scalings. In principle, the color correlation is proportional to this probability further conditioned on the probability that the height between the site in question and the boundary is non-zero at all $r$. However, unlike the argument for the $q < 1$ case, $p_0(r)$ is a small value for all $r' < r$ larger than a finite $r_0$, making the neglected prefactors of $1 - p_0(r)$ close to 1. They would alter the exact value of $\eta$, but cannot change the polynomial decay to exponential.

Table 5: Scaling of the correlation function $G_c(r)$ in 1D and 2D for different values of the deformation parameter $q$.

|  |  | 1D | 2D | $q$ |
|---|---|---|---|---|
|  |  | $e^{\alpha' r}$ | $e^{\beta' r^2}$ | $q > 1$ |
| $G_c(r) \sim$ |  | $r^{-\frac{3}{2}}$ | $r^{-\eta}$ | $q = 1$ |
|  |  | $e^{-\alpha r}$ | $e^{-\beta r}$ | $q < 1$ |

Table 6: Scaling of the spectral gap $\Delta_c(L)$ in 1D and 2D for different values of the deformation parameter $q$.

|  | 1D | 2D | $q$ |
|---|---|---|---|
|  | $< e^{-C_1 L^2}$ [17, 45] | $< e^{-C_2 L^3}$ [18] | $q > 1$ |
| $\Delta_c(L)$ | $\sim L^{-C}$ [12] | **Conjecture:** $\sim L^{-C'}$ | $q = 1$ |
|  | $\sim$ const. [44] | **Conjecture:** $\sim$ const. | $q < 1$ |

For the $q > 1$ phase, it is obvious that the color correlation grows with distance. In fact even exponentially. This is because for every configuration where the nearest down spin at the same height is at $r$, one can find another configuration where the nearest down spin at the same height is at $r + 1$. The latter configurations is weighed by an extra weight factor of $q^{r^d}$ in the GS superposition, where $d$ is the dimension of the model, since its entire random surface is shifted one step upward and thus have a larger volume beneath. Of course, the probability that gives the color correlation is normalized to 1, so one do not need to worry about diverging correlations for the bounded system, which the special entanglement and correlation properties of the models are built upon.

The above scaling behaviors can be summarized by Table. 5. The color correlation scaling is to be compared with the decay of spectral gap $\Delta_c(L)$ in the color sector, given in Table. 6.

# 6 Conclusion

We constructed exact 3D TN representations of the first examples of a 2D GSs with extensive EE. They closely mimic and directly generalize the 2D TN representation of the extensively entangled 1D GS. Like their 2D counterpart, our 3D TNs feature a holographic bulk, with the physical degrees of freedom located on the boundary. The 3D bulk allows those degrees of freedom to be connected in arbitrarily distant pairs on the lattice, such that bipartite EE can scale extensively with the size of a subsystem. Unlike the 1D TN, which consists entirely of the same tensor, the 2D networks consist of two types of tensors, one for each sublattice, as the 2D systems are entangled in multiple directions. Both types of tensors multitask to carry entanglement in different directions, reflecting the coupling between neighboring lattice sites belonging to the two different sublattices. Our TNs provide explicit demonstrations of classical descriptions of entangled quantum states, as the TNs are constructed from 3D tilings of tiles that correspond to non-vanishing entries of the tensors, which are shown to have a bijection to the 2D physical configurations in the GS superposition. We believe our constructions reveal new understanding of entanglement in 2D systems, and provides a new method for further generalizations to higher dimensions.

Independent of the TN constructed here, we have also made analytical arguments for the order parameter in the spin sector and the correlation function in the color sector. The spatially varying spin order parameter can be understood to some extent as the spin correlation with the boundary spin, the decaying behavior of which would then agree with the universal features of a phase transition from ordered to disordered phase. This is because the exotic order in this model is largely attributed to the presence of the boundary condition. The correlation function in the color sector, which is responsible for the exotic entanglement, has anomalous behavior in the extensively entangled phase. Some of the arguments for the correlation decays are based on the continuous field theory description of random surfaces in the scaling limit, especially in the 2D case. One possible future direction is to use the exact TN representation to either numerically confirm or algebraically derive the correlation functions.

# Acknowledgements

ZZ thanks Israel Klich, Robert Seiringer and Norbert Schuch for discussions.

# A Parent Hamiltonians of the highly entangled GSs

## A.1 The Fredkin spin chain

The Hamiltonian $H_F(q)$ of the colorful deformed Fredkin spin chain is given by

$$H_F(q) = H_\partial + H_C + \sum_{i=1}^{N-2} H_i(q). \tag{A.1}$$

The boundary term $H_\partial$ is given by

$$H_\partial = \sum_{c=1}^{s} \left( |\downarrow^c\rangle_0 \langle \downarrow^c | + |\uparrow^c\rangle_{N-1} \langle \uparrow^c | \right), \tag{A.2}$$

where the sum over $c$ is over $s$ different colors. The boundary term gives any spin configuration with a down spin at $i = 0$ and/or an up spin at $i = N-1$, an energy contribution. The term $H_C$ is the color mixing term, and is given by

$$
H_C = \sum_{i=0}^{N-2} \left[ \sum_{\substack{c_1,c_2=1 \\ c_1 \neq c_2}}^{s} |... \uparrow_i^{c_1} \downarrow_{i+1}^{c_2} ...\rangle \langle ... \downarrow_{i+1}^{c_2} \uparrow_i^{c_1} ...| \right]
$$
$$
+ \sum_{i=0}^{N-2} \frac{1}{2} \left[ \sum_{c_1,c_2=1}^{s} \left( |... \uparrow_i^{c_1} \downarrow_{i+1}^{c_1} ...\rangle - |... \uparrow_i^{c_2} \downarrow_{i+1}^{c_2} ...\rangle \right) \left( \langle ... \downarrow_{i+1}^{c_1} \uparrow_i^{c_1} ...| - \langle ... \downarrow_{i+1}^{c_2} \uparrow_i^{c_2} ...| \right) \right]. \tag{A.3}
$$

The bulk terms $H_i(t)$ are given in terms of projectors onto states $|F_{i,1}^{c_1,c_2,c_3}\rangle$ and $|F_{i,2}^{c_1,c_2,c_3}\rangle$ as

$$H_i(q) = \sum_{c_1,c_2,c_3=1}^{s} \left( |F_{i,1}^{c_1,c_2,c_3}\rangle \langle F_{i,1}^{c_1,c_2,c_3}| + |F_{i,2}^{c_1,c_2,c_3}\rangle \langle F_{i,2}^{c_1,c_2,c_3}| \right). \tag{A.4}$$

where the states in the projectors are given by

$$
\begin{aligned}
|F_{i,1}^{c_1,c_2,c_3}\rangle &= \frac{1}{\sqrt{q^{-2}+q^2}} \left( q^{-1} |... \uparrow_{i-1}^{c_1} \uparrow_i^{c_2} \downarrow_{i+1}^{c_3} ...\rangle - q |... \uparrow_{i-1}^{c_2} \downarrow_i^{c_3} \uparrow_{i+1}^{c_1} ...\rangle \right), \\
|F_{i,2}^{c_1,c_2,c_3}\rangle &= \frac{1}{\sqrt{q^{-2}+q^2}} \left( q^{-1} |... \uparrow_{i-1}^{c_1} \downarrow_i^{c_2} \downarrow_{i+1}^{c_3} ...\rangle - q |... \downarrow_{i-1}^{c_3} \uparrow_i^{c_1} \downarrow_{i+1}^{c_2} ...\rangle \right).
\end{aligned} \tag{A.5}
$$

To show that the state in Eq. (2.2) is the unique GS of Eq. (A.1), we define the so-called Fredkin moves $F_1^i$ and $F_2^i$ at a site $i$, introduced in [13]. The Fredkin move $F_j^i$ relate the spin configurations in the expression for $|F_{i,j}^{c_1,c_2,c_3}\rangle$ in Eq. (A.5) for $j = 1,2$, and are graphically seen in Fig. (17). To construct a state that is annihilated by all bulk terms $H_i(q)$, one can start from a given spin configuration and then include spin configurations related by a Fredkin move at $i$ in the state. By doing this for all $i$ and for all included spin configurations, we can assure that each $H_i(q)$ is annihilated by weighing the spin configurations correctly. The correct weighting can be determined by looking at Fig. (17), where we see that Fredkin moves relate spin configurations that differ in 2 units of area under the line segments traced out by the walk. In Eq. (A.5), we have weighted the low area configuration with a factor $q^2$ more than the high area configuration. That is, the low area configuration is weighted a factor $q^{\Delta A}$ more, where $\Delta A$ is the difference in area under

the walks. This implies that the correct weighting of the configurations for the projector $H_i(q)$ to be annihilated is to weigh the high area configuration a factor of $q^{\Delta A}$ more than the low area configuration. This is achieved by weighing each Dyck walk $w$ by a factor $q^{A(w)}$, where $A(w)$ is the area beneath the walk $w$. This is exactly what is done in the GS in Eq. (2.2).

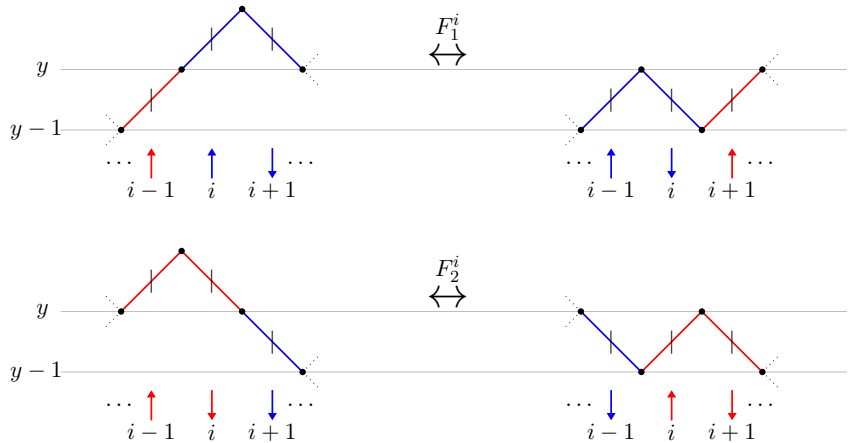

Figure 17: Figure depicting the two Fredkin moves $F_j^i$. The vertical black ticks on the Dyck walk segments indicate the position of the spins, $i-1$, $i$ and $i+1$, seen below. The black dots indicate positions in between where steps in the Dyck walk are taken from. The gray lines indicates the height $y$, and the area beneath the walk is reduced by 2 units when moving from left to right in the figures.

As noted in [13], by applying a Fredkin move $F_j^i$ to a spin configuration described by a Dyck walk, one always gets another Dyck walk. Starting from a spin configuration and including all spin configurations related by a series of Fredkin moves, will lead to a state annihilated by all $H_i(q)$, but only if the initial spin configuration is described by a Dyck walk will this procedure lead to a state that is annihilated by the boundary term $H_\partial$. For example, if we start from a spin configuration described by a walk that reaches negative height at some point, successive Fredkin moves will show that we are forced to include a spin configuration where the spin at $i = 0$ is a down spin in order to make sure that the projector $H_{i=1}(q)$ is annihilated. Similarly, if we start from a state described by a walk that ends at a positive height, successive Fredkin moves shows that we are forced to include a state with an up spin at $i = N - 1$ in order to make sure $H_{i=N-2}(q)$ is annihilated.

We note that the Fredkin moves in Fig. (17) always preserve the color correlation in the walks, meaning that color correlated Dyck walks are always taken to color correlated Dyck walks by applying a Fredkin move. Thus, the first sum of projectors in Eq. (A.3), which penalize walks where colors are not correlated, is annihilated by the state constructed by the procedure outlined above. The last sum of projectors in Eq. (A.3) is annihilated as long as we include an even superposition over correlated colors.

To understand the unusual entanglement properties of the state in Eq. (2.2), one can perform a Schmidt decomposition at the midpoint of the spin chain. Following, [16] this gives

$$|\text{GS}(q)\rangle = \sum_{m=1}^{(2s)^N} \alpha_m |\psi_m^{0,...,N/2-1}\rangle \otimes |\psi_m^{N/2,...,N-1}\rangle,$$

$$= \sum_{m=0}^{N/2} \sqrt{p_{N/2,m}(s,q)} \sum_{x\in\{\uparrow^1,\uparrow^2,...,\uparrow^s\}^m} |C_{0,m,x}^{0,...,N/2-1}\rangle \otimes |C_{m,0,\bar{x}}^{N/2,...,N-1}\rangle. \tag{A.6}$$

We have used that $|\text{GS}(q)\rangle$ is a superposition of color correlated Dyck walks, to obtain the second expression. Now, the states $|C_{a,b,x}^{i,...,j}\rangle$, are area weighted superpositions of spin configurations with $a$ unmatched down arrows and $b$ unmatched up arrows, and a specific coloring $x$ of the unmatched arrows. Since we know that every Dyck walk in $|\text{GS}(q)\rangle$ is color correlated, we know that if the walk in the first half of the chain have coloring $x$ of unmatched arrows, the state in the other half must have the matching coloring of $\bar{x}$. In [16] and [17], an analysis of the Schmidt coefficients reveals that the ground state undergoes a quantum phase transition at the $q = 1$ point, from an entanglement entropy that satisfies the area law, to extensive entanglement entropy, for the colored Fredkin spin chain ($s \geq 2$). At the critical point $q = 1$, the entanglement entropy goes as $\mathcal{O}(\sqrt{N})$, as was obtained in [13]. For the uncolored case, $s = 1$, the ground state satisfies an area law both for the $q > 1$ and $q < 1$ case. These results are summarized in the phase diagram in Fig. (18). This result matches exactly those found for the Motzkin spin chain in [12] and [15], and an identical figure is seen in [15].

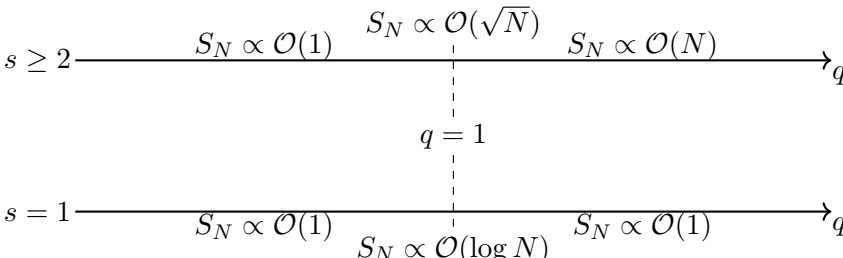

Figure 18: Phase diagram of the Fredkin spin chain, showing how the entanglement entropy $S_N$ for the half chain partition of a colored ($s \geq 2$) Fredkin spin chain of length $N$, undergoes a phase transition at the critical point $q = 1$. For $q > 1$ the entanglement entropy is extensive, while for $q < 1$ it is bounded by a constant. For the uncolored case $s = 1$ the entanglement entropy is bounded for both $q > 1$ and $q < 1$, but shows a logarithmic violation of the area law at $q = 1$.

## A.2   Fredkin chains coupled by 6-vertex rules on a square lattice

The Hamiltonian $H_{6\text{-vert}}$ of the coupled Fredkin chains with 6-vertex rules is given by

$$H_{6\text{-vert}} = H_0 + H_\partial + H_S(q) + H_C. \tag{A.7}$$

Firstly, we have $H_0$, the so-called "ice-rule" term, which couples spins at the vertices of the lattice. It demands that in each spin configuration in the GS of $H_{6\text{-vert}}$, there must be two arrows pointing into and two arrows pointing out from each ice rule vertex. For each vertex there is six such configurations, all seen in panel (b) in Fig. (1), giving the model its name. Following [18], we write $H_0$ for a square lattice of linear size $L$ as

$$H_0 = \sum_{x,y=0}^{L-1} H_{x,y}^0 = \sum_{x,y=0}^{L-1} \left( S_{x,y-1}^h - S_{x,y}^h - S_{x-1,y}^v + S_{x,y}^v \right)^2. \tag{A.8}$$

Spin configurations satisfying the ice-rule can be described by a well-defined height function $\phi$ on the dual lattice, defined in Eq. (2.3). By setting the zero point $\phi = 0$, the rules in Eq. (2.3) uniquely fixes the height function at the rest of the dual lattice points. We note that in Eq. (A.8), we consider ice rule vertices that involve spins outside our system, such as at $(x, y) = (-1, 0)$. These spins need to be considered in order for the height to be properly defined inside our system, and will be fixed by the boundary term $H_\partial$. The boundary term is given by

$$
\begin{aligned}
H_\partial = \sum_{y=0}^{L-2}(S_{L-1,y}^h - S_{0,y}^h) + \sum_{x=1}^{L-2}(-1)^{x+1}(S_{x,0}^h + S_{x,L-2}^h) \\
+ \sum_{x=0}^{L-2}(S_{x,L-1}^v - S_{x,0}^h) + \sum_{y=1}^{L-2}(-1)^{y+1}(S_{0,y}^v + S_{L-2,y}^v) + 4L - 6.
\end{aligned}
\tag{A.9}
$$

This specific form of the boundary term fixes several of the spins in the system. Firstly, it fixes the horizontal (resp. vertical) spins on the boundaries defined by $x = 0$ and $x = L-1$ (resp. $y = 0$ and $y = L - 1$), to the configuration seen in panel (a) in Fig. (1). Secondly, it also fixes the $y = 0$ and $y = L - 2$ (resp. $x = 0$ and $x = L - 2$) rows of horizontal (resp. vertical) spins, in the manner seen in the figure. It is only this exact configuration that exactly cancels the constant of $4L - 6$ in Eq. (A.9). Finally, together with the ice-rule term $H_0$, this specific boundary term fixes the gray spins outside our system, to the configuration seen in panel (a) in Fig. (1). This entails the alternating height at the height vertices outside our system, and implies that the even numbered spin chains in our system are raised one unit of height compared to the odd numbered spin chains. The black spins seen in the figure, are virtual spins raising the even numbered spin chains one height unit. Note that we have defined the zero point of the height at vertex $(x, y) = (-\frac{1}{2}, -\frac{1}{2})$.

The correlated swapping term $H_S(q)$ in Eq. (A.7), is the analog of the the sum of the $H_i(q)$ terms in Eq. (A.1). It can be written in terms of states $|F_{x,y,j}^{c_1,...,c_6}\rangle$, which are superpositions of spin configurations related by one of the four 2D Fredkin moves $F_{x,y,j}^{c_1,...,c_6}$, seen in Fig. (19).

By applying a 2D Fredkin move to a spin configuration that satisfies the ice rule at all ice rule vertices, we always get another spin configuration that satisfies the ice rule at all ice rule vertices. And conversely, by applying the moves to a spin configuration that does not satisfy the ice rule at all ice rule vertices, we get another spin configuration that does not satisfy the ice rule at all ice rule vertices. This is seen by noting that at each ice rule vertex in the spin configuration related by a move $F_{x,y,j}^{c_1,...,c_6}$, seen in Fig. (19), the same number of arrows is pointing in and out. The same exact statement also applies to spin configurations that have non-negative height at all height vertices. That is, by applying the 2D Fredkin move to a spin configuration of non-negative height at all height vertices, we get another spin configuration of non-negative height at all height vertices. This is seen by looking at the height of the height vertices in the spin configurations related by $F_{x,y,j}^{c_1,...,c_6}$. These two properties should be compared to the properties of the Fredkin moves of the previous chapter, seen in Fig. (17). We can then write $H_S(q)$ as

$$
H_S(q) = \sum_{x,y=1}^{L-2} \sum_{j=1}^{4} \sum_{c_1,...,c_6=1}^{s} |F_{x,y,j}^{c_1,...,c_6}\rangle\langle F_{x,y,j}^{c_1,...,c_6}|,
\tag{A.10}
$$

where the state $|F_{x,y,j}^{c_1,...,c_6}\rangle$ is defined as

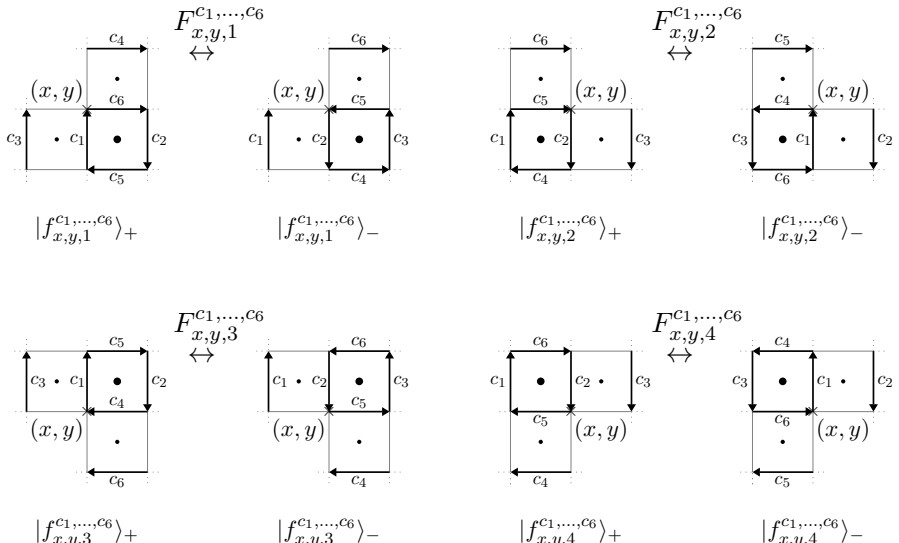

Figure 19: Figure showing the four different 2D Fredkin moves $F_{x,y,j}^{c_1,\ldots,c_6}$, and how they relate states $|f_{x,y,j}^{c_1,\ldots,c_6}\rangle_\pm$. The black dots mark height vertices, and the height vertex marked with the larger dot is reduced with 2 moving from left to right within a Fredkin move. The other height vertices remain at the same height. This is seen from Eq. (2.3). The $c_i$ labels denote the colors of the spin.

$$|F_{x,y,j}^{c_1,\ldots,c_6}\rangle = \frac{1}{\sqrt{q^{-2}+q^2}}\left(q^{-1}|f_{x,y,j}^{c_1,\ldots,c_6}\rangle_+ - q|f_{x,y,j}^{c_1,\ldots,c_6}\rangle_-\right), \qquad (A.11)$$

where the states $|f_{x,y,j}^{c_1,\ldots,c_6}\rangle_\pm$ are defined in Fig. (19), for $j = 1, 2, 3, 4$. The correlated swapping term $H_S(q)$ acts as the term term $H_i(q)$ in Eq. (A.1), for both the horizontal and vertical spin chains, which can be easily seen by ignoring either the horizontal or vertical spins in Fig. (19), and then compare with Fig. (17). This will ensure that the GS will consist of spin configurations, which are described by Dyck walks in all horizontal and vertical spin chains. By tuning the deformation parameter $q$ to large values, we can ensure that spin configurations with high Dyck walks is weighted more in the GS superposition.

It is clear that the spin configurations on the left hand side of each Fredkin move in Fig. (19), have a volume 2 more than the spin configurations on the right hand side. In analogy with Eq. (2.2), the spin configurations $S$ should be weighed by $q^{V(S)}$ in the GS, where $V(S)$ is the volume of the spin configuration defined in Eq. (2.4).

Finally, we have the color mixing term, $H_C$, given by

$$
\begin{aligned}
H_C = &\sum_{\substack{x,y=0 \\ }}^{L-2} \sum_{\substack{c_1,c_2=1 \\ c_1 \neq c_2}}^{s} \left(|\uparrow_x^{c_1}\downarrow_{x+1}^{c_2}\rangle_y\langle\downarrow_{x+1}^{c_2}\uparrow_x^{c_1}| + |\,{}_{c_1}^{c_2}\!\!\overset{\rightarrow y+1}{\underset{\leftarrow y}{}}\rangle_x\langle\,{}_{c_1}^{c_2}\!\!\overset{\rightarrow y+1}{\underset{\leftarrow y}{}}|\right) \\
&+ \sum_{x,y=0}^{L-2}\sum_{c_1,c_2=1}^{s}\left(|\uparrow_x^{c_1}\downarrow_{x+1}^{c_1}\rangle_y - |\uparrow_x^{c_2}\downarrow_{x+1}^{c_2}\rangle_y\right)\left(\langle\downarrow_{x+1}^{c_1}\uparrow_x^{c_1}| - \langle\downarrow_{x+1}^{c_2}\uparrow_x^{c_2}|\right) \\
&+ \sum_{x,y=0}^{L-2}\sum_{c_1,c_2=1}^{s}\left(|\,{}_{c_1}^{c_1}\!\!\overset{\rightarrow y+1}{\underset{\leftarrow y}{}}\rangle_x - |\,{}_{c_2}^{c_2}\!\!\overset{\rightarrow y+1}{\underset{\leftarrow y}{}}\rangle_x\right)\left(\langle\,{}_{c_1}^{c_1}\!\!\overset{\rightarrow y+1}{\underset{\leftarrow y}{}}| - \langle\,{}_{c_2}^{c_2}\!\!\overset{\rightarrow y+1}{\underset{\leftarrow y}{}}|\right),
\end{aligned}
$$
$$(A.12)$$

The $H_C$ term serves the same purpose as $H_C$ in Eq. (A.3), namely to ensure that all spin configurations present in the ground state superposition are described by superpositions of color correlated Dyck walks. The first term in Eq. (A.12) penalize spin configurations that are not described by color correlated Dyck walks in the horizontal and vertical spin chains. The last two terms ensure that we get a superposition of valid colorings in the ground state. In combination with the correlated swapping term $H_C$, spin configurations where the color of a horizontal (resp. vertical) down spin, does not match that of the closest up spin in the same sublattice to the left (resp. beneath), gets penalized no matter how far separated the unmatched spins are.

$$s \geq 2 \xrightarrow[\hspace{2cm}]{\begin{array}{ccc} S_L \propto \mathcal{O}(L) & \begin{array}{c} S_L \propto \mathcal{O}(L \log L) \end{array} & S_L \propto \mathcal{O}(L^2) \end{array}} q$$

$$q = 1$$

Figure 20: Phase diagram of the model, showing how the entanglement entropy $S_L$ between a bipartition of the colored ($s \geq 2$) 6-vertex model of size $L \times L$, undergoes a phase transition at the critical point $q = 1$. For $q > 1$ the entanglement entropy is extensive, while for $q < 1$ it is bounded by the size of the boundary between the bipartition.

Due to the rotational symmetry of the lattice, the EE between subsystems has the same scaling for any bipartition through the center of the system. After a careful analysis of the Schmidt coefficients, one obtains the phase diagram seen in Fig. (20) [18]. Again the leading contribution to the EE is due to the combination of correlations in the color degree of freedom and the dominating presence of high volume spin configurations in Eq. (2.5).

## A.3 Fredkin chains coupled by lozenge tiling rules on a triangular lattice

The full Hamiltonian $H_{\text{loz}}$ of the lozenge tiling model is given by

$$H_{\text{loz}} = H_0 + H_\partial + H_S + H_C. \tag{A.13}$$

In the GS of Eq. (A.13), we have the requirement that each triangular face of the triangular lattice $\Lambda$ is covered exactly once by lozenge. This is equivalent to requiring that the each lattice point on the hexagonal dual lattice $\Lambda^*$ is covered exactly once by a dimer. In [19], this property is ensured in the GS by the term $H_0$, given by

$$H_0 = \sum_{\boldsymbol{r} \in \Lambda^*} \sum_{i=1}^{3} (n_{\boldsymbol{r}, \boldsymbol{r}+\boldsymbol{e}_i} - 1)^2, \tag{A.14}$$

where $n_{\boldsymbol{r}, \boldsymbol{r}+\boldsymbol{e}_i}$ is the number operator for dimers covering lattice point $\boldsymbol{r}$ and $\boldsymbol{r} + \boldsymbol{e}_i$ on the dual hexagonal lattice $\Lambda^*$. To ensure that this requirement can be met at every $\boldsymbol{r}$ it is necessary to choose a domain $\mathcal{R}$ of the triangular lattice $\Lambda$ which is tileable by lozenges [34]. Such domains $\mathcal{R}$ satisfy the constraint in Eq. (2.7). Following, [19] we will only consider domains satisfying the stronger constraint, where $d(u, v)$ replaced by 1 in Eq. (2.7). As shown in Appendix D, domains $\mathcal{R}$ satisfying this stronger constraint are also *uniquely* tileable by hexagons. The lozenge tilings in Fig. (2) shows tilings on such a domain.

To ensure that the GS is a superposition of all tiling configurations that correspond to a random surface above the zero height plane in the scaling limit, an appropriate boundary term $H_\partial$ and a correlated swapping term $H_S$ is defined in [19]. The boundary term should penalize tilings with edges $u - v$ immediately inside the boundary at negative

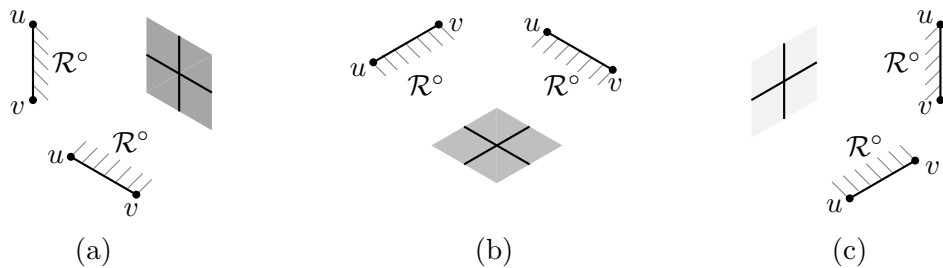

Figure 21: Panels (a) - (c) show the six different boundary edges $u - v$ of the triangular lattice, and the corresponding lozenge that both covers the triangular face and ensures that the height of the interior edges are non-zero. $\mathcal{R}^\circ$ marks the interior.

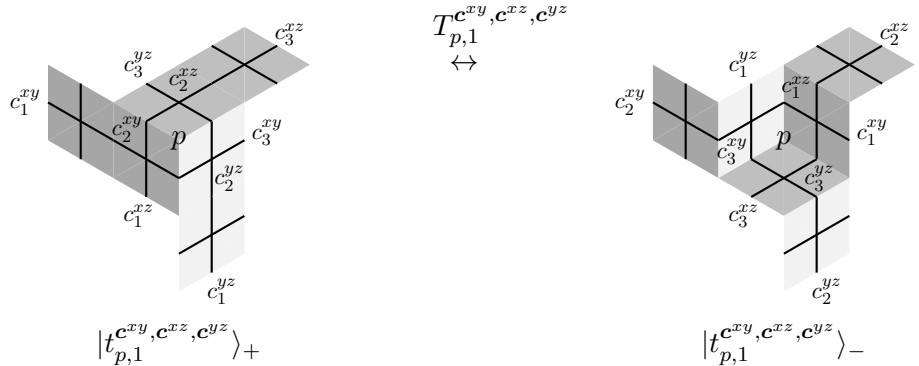

Figure 22: One of the 8 lozenge Fredkin moves acting on a 6-lozenge neighborhood $p$, denoted $T_{p,1}^{\boldsymbol{c}^{xy},\boldsymbol{c}^{xz},\boldsymbol{c}^{yz}}$. Note that moving from left to right reduces the volume by 3 and that a (resp. non) color correlated Dyck walk configuration will be sent to a (resp. non) color correlated Dyck walk configuration.

height. For domains of the triangular lattice there are only three possible orientations of edges, one for each of the axis in panel (a) in Fig. (2). For each of the three edges on the boundary, the interior of the domain $\mathcal{R}$ can be on either side, which in total gives six possible boundary segments. For each of these six boundary segments, there is only one lozenge that both covers the triangular face with the given boundary segment, and ensures that the height of the interior edges are non-zero. The six different boundary edges along with the corresponding lozenge are seen in Fig. (21). We denote the lozenge corresponding to the given boundary edge $u - v$ by $L_{u-v}$. Then we can define the boundary term $H_\partial$ as

$$H_\partial = \sum_{u-v \in \partial \mathcal{R}} \left( \hat{1} - |L_{u-v}^s\rangle\langle L_{u-v}^s| \right), \tag{A.15}$$

where $|L_{u-v}^s\rangle$ the state corresponding to the appropriate lozenge for the given boundary edge $u - v$, summed over the $s$ different colorings. The boundary term fixes the boundary lozenges to a specific lozenge, in the same way the boundary term in the previous sections fixes the boundary spins. Note that the definition of the boundary term in Eq. (A.15) differs slightly from, but is ultimately equivalent to, the definition given in [19].

Next, the swapping term $H_S$ is defined in such a way that it works together with $H_\partial$ to guarantee that by including any lozenge tiling where the height at any edge $u - v$ in the interior is negative (or equivalently: that the height at any height vertex is less tham $-1/2$), will be penalized with a positive energy. In addition, it also ensures that

the tilings with large volume dominate the ground state superposition for deformation parameter $q > 1$. It is defined in terms of projectors onto states $|T_{p,j}^{\boldsymbol{c}^{xy},\boldsymbol{c}^{xz},\boldsymbol{c}^{yz}}\rangle$, which are volume weighted superpositions between lozenge tilings related by a conditioned lozenge move $T_{p,j}^{\boldsymbol{c}^{xy},\boldsymbol{c}^{xz},\boldsymbol{c}^{yz}}$. These moves are the lozenge tiling analogue of the Fredkin moves in Fig. (19). There are 8 such moves, which can be seen as Eq. (4) in [19], and one of them is drawn in Fig. (22). In this notation, $p$ denotes a 6-lozenge neighborhood, and $\boldsymbol{c}^{ij} = \begin{pmatrix} c_1^{ij} & c_2^{ij} & c_3^{ij} \end{pmatrix}$ are vectors containing the color labels of the lines in the $ij$-plane of a lozenge. Applying one of the lozenge Fredkin moves to a lozenge tiling which is described by color correlated Dyck walks in the $xy$-, $xz$- and $yz$-plane, gives another such lozenge tiling. That can be seen for each of the moves in Eq. (4) in [19], and for one of the moves in Fig. (22). We can then write the correlated swapping term $H_{\mathrm{S}}$ as

$$H_{\mathrm{S}} = \sum_{p \in \mathcal{R}_6^{\circ}} \sum_{\{\boldsymbol{c}^{xy},\boldsymbol{c}^{xz},\boldsymbol{c}^{yz}\}} \sum_{j=1}^{8} |T_{p,j}^{\boldsymbol{c}^{xy},\boldsymbol{c}^{xz},\boldsymbol{c}^{yz}}\rangle\langle T_{p,j}^{\boldsymbol{c}^{xy},\boldsymbol{c}^{xz},\boldsymbol{c}^{yz}}|, \tag{A.16}$$

where $\mathcal{R}_6^{\circ}$ denotes all 6-lozenge neighborhoods in the bulk of the domain $\mathcal{R}$. The state $|T_{p,j}^{\boldsymbol{c}^{xy},\boldsymbol{c}^{xz},\boldsymbol{c}^{yz}}\rangle$ is defined as

$$|T_{p,j}^{\boldsymbol{c}^{xy},\boldsymbol{c}^{xz},\boldsymbol{c}^{yz}}\rangle = \frac{1}{\sqrt{q^{-3/2} + q^{3/2}}} \left( q^{-3/2}|t_{p,j}^{\boldsymbol{c}^{xy},\boldsymbol{c}^{xz},\boldsymbol{c}^{yz}}\rangle_{+} - q^{3/2}|t_{p,j}^{\boldsymbol{c}^{xy},\boldsymbol{c}^{xz},\boldsymbol{c}^{yz}}\rangle_{-} \right), \tag{A.17}$$

where the states $|t_{p,j}^{\boldsymbol{c}^{xy},\boldsymbol{c}^{xz},\boldsymbol{c}^{yz}}\rangle_{\pm}$ are the lozenge analogue of the states $|f_{x,y,j}^{c_1,...,c_6}\rangle_{\pm}$ defined in Fig. (19). The states are related by a lozenge Fredkin move, see Fig. (22) for an example. Now, we define the volume of a given lozenge tiling $T$ as

$$V(T) = \sum_{u \in \mathcal{R}} \phi_u \tag{A.18}$$

that is the volume $V(T)$ is the sum of the height at all height vertices. The volume of tilings described by the states $|t_{p,j}^{\boldsymbol{c}^{xy},\boldsymbol{c}^{xz},\boldsymbol{c}^{yz}}\rangle_{-}$ is lower by 3 volume units than the volume of tilings described by the states $|t_{p,j}^{\boldsymbol{c}^{xy},\boldsymbol{c}^{xz},\boldsymbol{c}^{yz}}\rangle_{+}$. This is seen by inspecting Eq. (4) in [19], and can be seen for one of the moves in Fig. (22). Since the low volume states in Eq. (A.17) are weighted a factor $q^3$ more than the high volume states, the correct weighing of lozenge tilings $T$ in the GS (2.8) becomes $q^{V(q)}$.

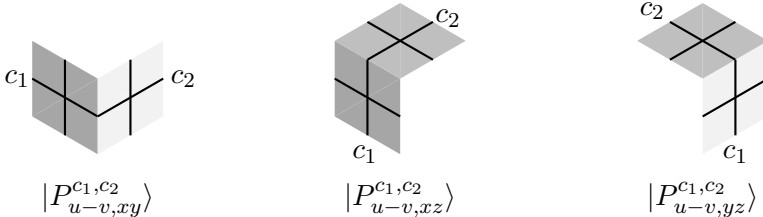

Figure 23: The states $|P_{u-v,ij}^{c_1,c_2}\rangle$, ensuring a superposition of color correlated Dyck walks in the three planes in the term $H_{\mathrm{C}}$ in Eq. (A.19).

Finally, the appropriate color mixing term $H_{\mathrm{C}}$ is introduced. As usual it ensures two things. Firstly, that the Dyck walks in the ground state are color correlated and secondly, that there is a superposition of color in the color correlation of the Dyck walks. In [19], this is achieved for the lozenge tiling model by defining $H_{\mathrm{C}}$ as

$$H_{\rm C} = \sum_{u-v\in\mathcal{R}^\circ} \sum_{ij\in\{xy,xz,yz\}} \sum_{c_1\neq c_2} [|P^{c_1,c_2}_{u-v,ij}\rangle\langle P^{c_1,c_2}_{u-v,ij}|$$
$$+ \frac{1}{2}\left(|P^{c_1,c_1}_{u-v,ij}\rangle - |P^{c_2,c_2}_{u-v,ij}\rangle\right)\left(\langle P^{c_1,c_1}_{u-v,ij}| - \langle P^{c_2,c_2}_{u-v,ij}|\right)], \tag{A.19}$$

where we sum over all edges $u-v$ in the bulk $\mathcal{R}^\circ$ and over all three planes $ij$. The states $|P^{c_1,c_2}_{u-v,ij}\rangle$ are seen in Fig. (23). The first term in Eq. (A.19) together with $H_{\rm C}$ ensures the proper color correlation in the Dyck walk in the three planes. The second term ensures that we get a superposition of these color correlations in the ground state.

## B  Bijection between 3D tilings of cubes and 6-vertex configurations

### B.1  Injection from cross-sections of valid 3D tilings to arrays of Dyck paths

The one-to-one correspondence between valid tiling and spin configurations is established using the height function of spin configurations

$$\phi^{\rm h(v)}_{x,y} = \frac{1}{2}\left(\phi_{x+1/2,y+1/2} + \phi_{x\mp1/2,y\pm1/2} - 1\right), \tag{B.1}$$

where $\phi^{\rm h(v)}_{x,y}$ is the height for horizontal (vertical) spins. Next, we note that each tower of cubes has exactly one $X_2$ or $X_4$ tile, for either $X = H, V$. Above these tiles we can only place the $X_3$ tile and below $X_2$ (resp. $X_4$) only $X_1$ (resp. $X_5$), for the faces of neighboring cubes to match. This is also the case for the $A_2$ and $A_4$ tiles for the single Fredkin chain. The $X_2$ or $X_4$ tile is placed at the level $l = h^{\rm h(v)}_{x,y}$ in the tower of tiles at $(x,y)$, where

$$h^{\rm h(v)}_{x,y} = \frac{L}{2} - \phi^{\rm h(v)}_{x,y}, \tag{B.2}$$

and $l = \frac{L}{2}$ is the highest level in the TN. We will refer to $h^{\rm h(v)}_{x,y}$ as the height of the arrowed path at $(x,y)$. It follows that the number of tiles $M_{x,y}$ in the tower of tiles at $(x,y)$ is determined by the range of possible values for $h^{\rm h(v)}_{x,y}$, which is larger in the middle of the system than near the boundaries. This confirms the inverse step pyramid shape of the network as expected. Note that the height of the four spins in each corner, is constrained to the value of 1 by the specific boundary conditions we have chosen. Therefore, the $X_2$ and $X_4$ tiles corresponding to these spins are always placed at level $l = \frac{L}{2} - 1$. This gives the four squares of 4 square tiles around the center in the $l = 1$ level in Fig. (7).

The tiles corresponding to individual horizontal and vertical spin chains in Fig. (7) matches the tiles used for the single Fredkin spin chain in Fig. (3). This is revealed by the cutouts of the valid tilings in Fig. (7) indicated by thick black lines, with a side-view shown in Fig. (8). In the vertical projection plane along the chains, a clear correspondence between the cubical tiles and the square tiles in Fig. (3) is observed: $X_i \leftrightarrow A_i$ with $X = H, V$ for horizontal and vertical spin chains respectively. Using this correspondence, we can argue that the cubical tiles incorporate the Dyck walk rule. This means that the set of cube tiles only admit valid tilings for spin configurations described by Dyck walks in all horizontal and vertical chains (but not for all such spin configurations as we soon shall

see). To support this claim, we extend the arguments from Ref. [28] for single Fredkin chain tiles to the cube tiles of each individual spin chain in our 2D system. Firstly, we note that by fixing the spins in a given spin chain, that is fixing the in- and outflow of arrows, the corresponding valid tiling for that spin chain is uniquely determined. Indeed, since none of the tiles is empty without arrows, or contains sources or drains of arrows, in a valid tiling the arrowed paths can only turn at the maximum height $h_{x,y}^{\mathrm{h(v)}}$, to avoid leaving empty tiles. Thus, they form "rainbows" of arrowed paths as seen in Fig. (3) and in Fig. (8).

We have shown that the mapping from the vertical cross-sections in both directions of valid tiling of cubes to the set of two separate arrays of Dyck paths is injective, meaning that cross-sections of all 3D tilings reproduces Dyck paths, and no two tilings give the same Dyck path configurations. However, spin configurations that are described by Dyck walks in all horizontal and vertical spin chains does not generally satisfy the ice rule at all vertices of the physical square lattice. As we know, the spin configurations $|S^C\rangle$ in Eq. (2.5) satisfy the ice rule at all ice rule vertices in addition to being described by Dyck walks. In order to show the one-to-one correspondence between spin configurations $|S^C\rangle$ in the GS and valid tilings of the cube tiles, we will now argue that the set of cube tiles only admit valid tilings for spin configurations compatible with the ice rule.

## B.2 Surjection from 4 neighboring towers of tiles to 6-vertex spin configurations

We now show that the cube tiles admit valid tilings only for spin configurations that satisfy the ice rule at all ice rule vertices. First, we show that there exist a single valid tiling for the six different vertex configurations allowed by the ice rule. To do this, we use that spin configurations satisfying the ice rule can be described by a well defined height function $\phi$, defined through the relations Eq. (2.3). We then insert the expression for $h_{x,y}^{\mathrm{h(v)}}$ in Eq. (B.2) for each spin around an ice rule vertex into Eq. (2.3), by using Eq. (B.1). This gives us the following relations for the heights of the arrowed paths around an ice rule vertex:

$$
\begin{aligned}
h_{x,y}^{\mathrm{h}} - h_{x,y}^{\mathrm{v}} &= S_{x,y}^{\mathrm{h}} - S_{x,y}^{\mathrm{v}}, \\
h_{x,y}^{\mathrm{h}} - h_{x-1,y}^{\mathrm{v}} &= -S_{x,y}^{\mathrm{h}} - S_{x-1,y}^{\mathrm{v}}, \\
h_{x,y-1}^{\mathrm{h}} - h_{x,y}^{\mathrm{v}} &= S_{x,y-1}^{\mathrm{h}} - S_{x,y}^{\mathrm{v}}, \\
h_{x,y-1}^{\mathrm{h}} - h_{x-1,y}^{\mathrm{v}} &= -S_{x,y-1}^{\mathrm{h}} + S_{x-1,y}^{\mathrm{v}}.
\end{aligned}
\tag{B.3}
$$

For each of the six vertices, we can insert the spin values in Eq. (B.3) and get the corresponding valid tiling seen in Fig. (24). For simplicity, we have relabeled coordinates as $(x,y) \leftrightarrow n$ (north) and $(x, y-1) \leftrightarrow s$ (south) for horizontal spins and $(x,y) \leftrightarrow e$ (east) and $(x-1,y) \leftrightarrow w$ (west) for vertical spins around an ice rule vertex. Note that in Fig. (24), all tiles at levels $l < l_1$ directly under the four tiles are either $X_1$ or $X_5$, and all tiles at levels $l > l_2$ directly above can only be the $X_3$ tile. For spin configurations satisfying the ice rule, the heights of the arrowed paths due to spins around an ice rule vertex are related by Eq. (B.3) for all ice rule vertices. Therefore, we know that the cube tiles are sufficient to produce valid tilings for all the six different vertices.

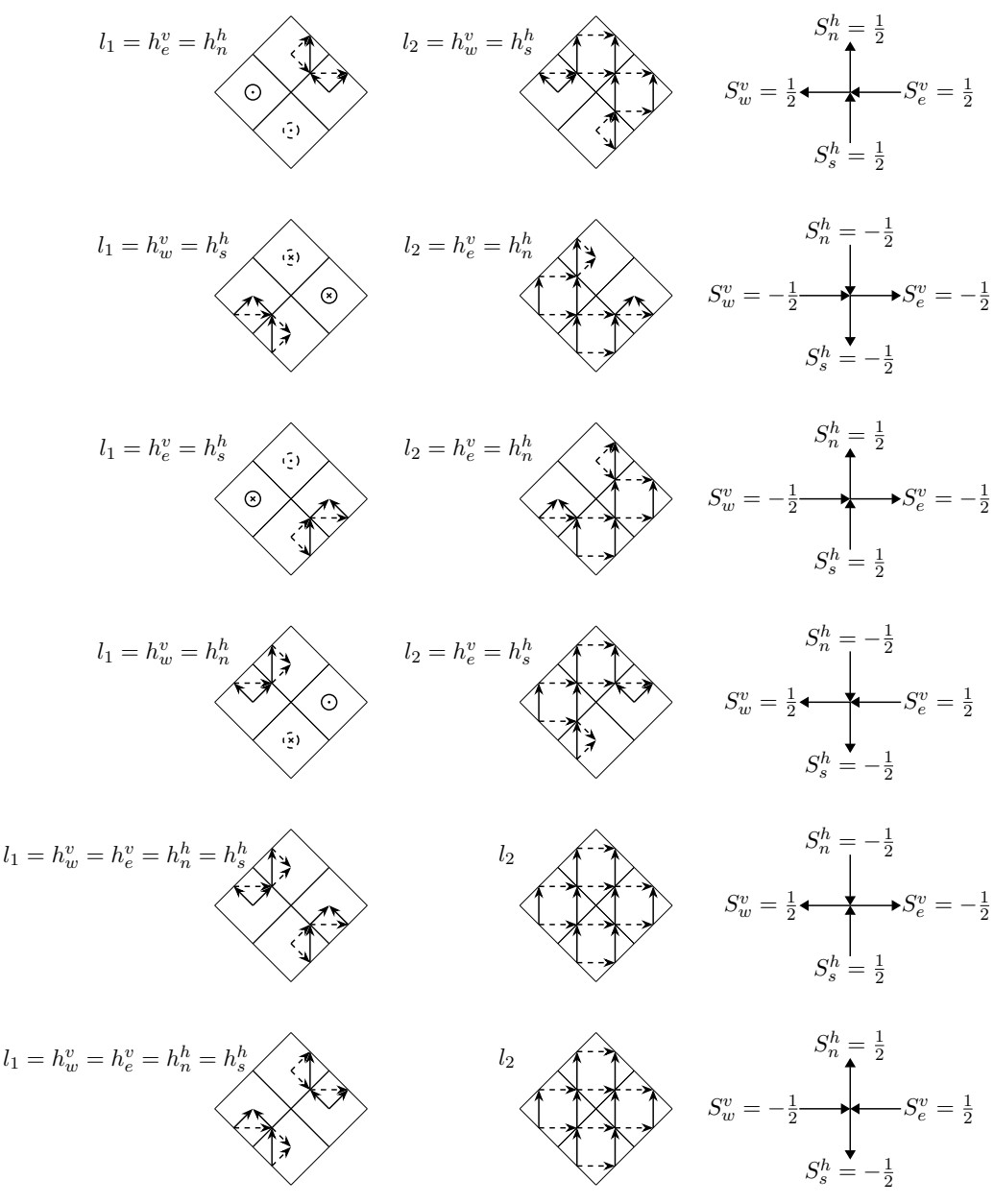

Figure 24: Figure showing the six vertices and their corresponding valid tilings. The leftmost column show tiles that will appear at a level $l = l_1$ in the valid tiling, while the middle column show tiles that will appear at $l = l_2 = l_1 + 1$. At the levels $l < l_1$ directly under the four tiles, only the $X_1$ or $X_5$ tiles appear. At the levels $l > l_2$ directly above the four tiles, only the $X_3$ tile appear.

Conversely, it is easy to convince oneself by inspection that the Fig. (24) contains all the valid tiling configurations involving a neighborhood of 4 tiles on a horizontal plane. This implies that all valid 3D tilings reproduce six-vertex configurations in the external legs. All the other 10 vertex configurations would violate the arrow continuity. An example of such is shown in Fig. (25). This completes the arguments that the mapping from valid 3D tilings of cubes to 6-vertex configurations of spins is surjective. Combined with the result of the previous section, it follows that valid 3D tilings are both Dyck (in two orthogonal sets of $xz$- and $yz$-cross-sections) and ice rule respecting (in the $xy$-plane). Since we have also shown that the mapping from tilings to Dyck paths is one-to-one, there is a one-to-

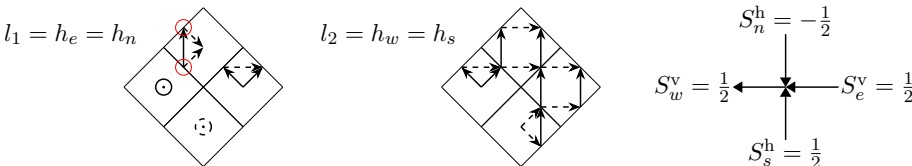

Figure 25: Figure showing at attempt of constructing a tiling corresponding to a spin configuration that violates the ice rule. The resulting violation of arrow continuity is marked by the red circles.

one mapping between valid 3D tilings and discrete 2D surfaces outlined by arrays of Dyck paths that are the physical configurations in $|S^C\rangle$ of Eq. (2.5).

## C    Determination of weighting for the 6-leg tensors

We now show how to obtain the weights of $q$ seen for the tiles in Table. 1. To obtain the weights, we follow the approach in Appendix F in Ref. [37] for an exact holographic TN representation of the GS of the uncolored Motzkin spin chain. Firstly, we rewrite the GS of the 6-vertex model for an arbitrary value of $q$ in terms of the GS at $q = 1$. It is understood that if we let all tiles have value 1, a superposition where all spin configurations are weighted equally is obtained by contracting the TN, which gives the GS in the $q = 1$ case. In the GS for an arbitrary $q$, given in Eq. (2.5), spin configurations are weighted by a factor $q^{V(S)}$. The volume $V(S)$ of a spin configuration is defined in terms of the height $\phi_{x+1/2,y+1/2}$ at the height vertices in Eq. (2.4). The height $\phi_{x+1/2,y+1/2}$ at a point $(x, y)$ of a spin configuration $|S^C\rangle$ in the GS can be expressed through the eigenvalue equation

$$H_{x,y}|S^C\rangle = \phi_{x+1/2,y+1/2}|S^C\rangle, \tag{C.1}$$

where $H_{x,y}$ is given by

$$
\begin{aligned}
H_{x,y} &= \sum_{x'=0}^{x} 2S^h_{x',y} + (x+1) \bmod 2, \\
&= \sum_{y'=0}^{y} 2S^v_{x,y'} + (y+1) \bmod 2, \\
&= \sum_{x'=0}^{x} S^h_{x',y} + \sum_{y'=0}^{y} S^v_{x,y'} + 0.5\left((x+1) \bmod 2 + (y+1) \bmod 2\right),
\end{aligned}
\tag{C.2}
$$

where we have used that the height can be expressed either in terms of the horizontal spin operators $S^h_{x,y}$ or the vertical spin operators $S^v_{x,y}$. Note that the modulo terms come from the fact that even numbered spin chains are raised one unit of height due to boundary conditions. If we now insert Eq. (C.2) into Eq. (C.1) and sum over $x$ and $y$ we obtain

$$\left[\frac{L}{2} + \sum_{x,y=0}^{L-2} (L-1-x)S^h_{x,y} + (L-1-y)S^v_{x,y}\right]|S^C\rangle = V(S^C)|S^C\rangle. \tag{C.3}$$

Note that the sum over $x$ and $y$ goes up until $L - 2$, as the height vertices furthest to the right (resp. furthest up) in our system is $\phi_{L-3/2,y}$ (resp. $\phi_{x,L-3/2}$). We can then write the

GS in Eq. (2.5) as

$$|\text{GS}(q)\rangle = \frac{1}{N} \sum_{S \in S_{L \times L}} {\sum_{C}}' q^{\frac{L}{2}} \prod_{x=0}^{L-2} \prod_{y=0}^{L-2} q^{(L-1-x)S_{x,y}^h + (L-1-y)S_{x,y}^v} |S^C\rangle,$$

$$= q^{\frac{L}{2}} \prod_{x=0}^{L-2} \prod_{y=0}^{L-2} q^{(L-1-x)S_{x,y}^h + (L-1-y)S_{x,y}^v} |\text{GS}(q=1)\rangle. \tag{C.4}$$

That is, in order to obtain the GS for an arbitrary $q$ we simply apply operators $q^{\alpha_x S_{x,y}^h}$ and $q^{\alpha_y S_{x,y}^v}$ to the physical legs of the TN, with $a_i = L - 1 - i$ for $i = x, y$. In addition, we multiply the state with a constant $q^{L/2}$. We now exchange the $q^{\alpha_x S_{x,y}^h}$ and $q^{\alpha_y S_{x,y}^v}$ operators applied to the physical legs with operators

$$\alpha_i = \text{diag}(q^{\frac{\alpha_i}{2}}, q^{\frac{\alpha_i}{4}}, 1, q^{\frac{-\alpha_i}{2}}) \tag{C.5}$$

with $i = x, y$ for horizontal and vertical spins respectively. The new operator is a $4 \times 4$ matrix, which takes the same value as $q^{\alpha_x S_{x,y}^h}$ and $q^{\alpha_y S_{x,y}^v}$, namely $q^{\frac{\pm \alpha_i}{2}}$, when indices are fixed to $(\pm 2 \boldsymbol{c}_1, \boldsymbol{0})$ for vertical spins and $(\boldsymbol{0}, \pm 2 \boldsymbol{c}_1)$ for horizontal spins. Since the physical legs of the tensor network only takes on these values, it is clear that applying the operators in Eq. (C.5) is equivalent to applying the operators $q^{\alpha_x S_{x,y}^h}$ and $q^{\alpha_y S_{x,y}^v}$ to the physical legs. The operator in Eq. (C.5) also takes on values $q^{\frac{\alpha_i}{4}}$ when its indices are fixed to $(\boldsymbol{c}_1, \boldsymbol{c}_2)$ and 1 when its indices are fixed to $(\boldsymbol{0}, \boldsymbol{0})$. These values of the operator is motivated by the U(1) invariance by our tensors, expressed by Eq. (3.2). By defining the operator as in Eq. (C.5), we have as a result of the U(1) invariance of the cube tiles, the following identity for the horizontal tensor $H$

$$\tag{C.6}$$

and the following identity for the vertical tensor $V$

$$\tag{C.7}$$

where we have used a square to represent the cube tensors and labeled the indices on which the operators are acting. Note that the $-\alpha$ on the right hand side of the equations, appear at the incoming indices of the dashed arrows (resp. solid arrows) for the $H$ tensor identity (resp. $V$ tensor identity). Also note that the operator marked by $\alpha_x$ (resp. $\alpha_y$) only act on dashed arrows (resp. solid arrows). The identities in Eq. (C.6) and Eq. (C.7) are seen

to hold for all horizontal and vertical tiles individually. Using the identities in Eq. (C.6) and Eq. (C.7) we can move the operators up in the tensor network until it reaches the top level. The operators will then be fixed to the value 1 by fixing of the $k_6$ legs of the tensor to the $(\mathbf{0}, \mathbf{0})$ value by the boundary tensor $\delta_{\mathbf{k}, (\mathbf{0}, \mathbf{0})}$. Then each level in the tensor network will look like panel (a) in Fig. (26).

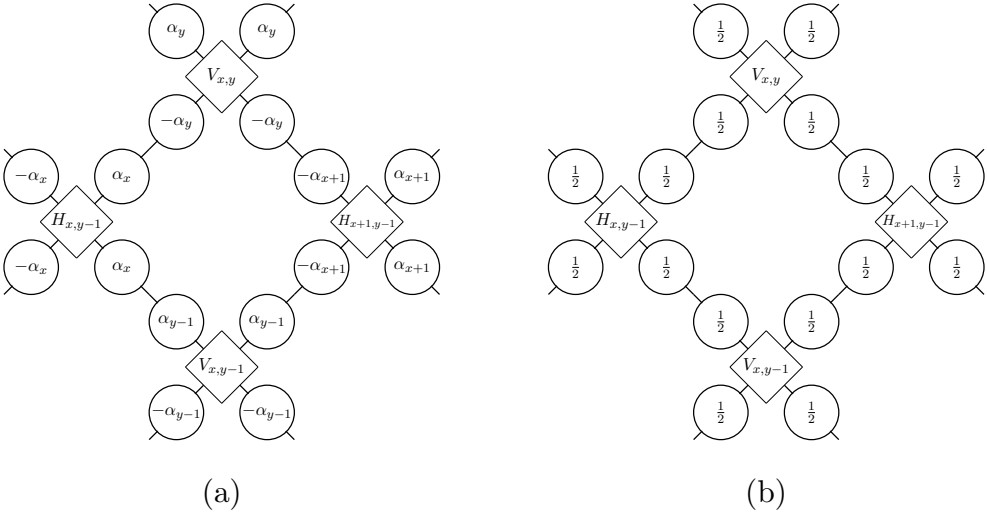

(a)                                                    (b)

Figure 26: (a) Showing how each level of the tensor network will look after having used the identities Eq. (C.6) and Eq. (C.7) to move the operators defined in Eq. (C.5) to the top level in the tensor network. (b) Figure showing how each level of the tensor network will look after having distributed the $\alpha_i$-tensors in (a).

To make further progress we use that we can interchange the order of the two operators between adjacent horizontal and vertical tensors, as the value of the various indices between adjacent tensors are of course identical since they are contracted. Then, we use that for each horizontal tensor, we can move $\alpha_x$-tensors contracted with the index $k_2$ to be contracted with $k_5$ index instead and vice versa, and similarly between $k_3$ and $k_4$. These pairs of indices are always equal for all $H_i$ tiles, as seen in Table. 1 which enables this maneuver. For the vertical tiles we can similarly move contractions with the $\alpha_y$-tensor between the indices $k_3$ and $k_5$, and between indices $k_2$ and $k_4$. By splitting each operator of value $\alpha_i$ into two copies of $\alpha_i/2$ and distributing these tensor according to the rules mentioned in the preceding sentences we arrive at panel (b) in Fig. (26). To obtain the results we have also used that $\alpha_i - \alpha_{i+1} = 1$ for $i = x, y$. By considering the contractions in panel (b) in Fig. (26) with the various tiles $X_i$, and using the definition of the tensor in Eq. (C.5) with $\alpha_i = 1/2$, we get the tile weights as seen in Table. 1.

# D    Bijection between 3D tilings of prisms and lozenge tilings

## D.1    Injectivity from 3D tilings to lozenge tilings with non-negative height

In this section we show that one can only tile valid tilings of the prism tiles corresponding to lozenge tilings of the three lozenges in Fig. (2), that are described by Dyck walks in all three planes. To show this, we use the height function $\phi$, which is defined on the lattice sites $u$ of the triangular lattice in Fig. (11). The spins, and the corresponding towers of

prism tiles, are centered around points of position $\boldsymbol{r}$, which are the lattice points of the dual hexagonal lattice. The position of the height vertices $u$ are therefore given by $\boldsymbol{r} + \boldsymbol{e}_i$ where $\boldsymbol{e}_i$ denotes a displacement vector in one of three directions. We define the height related to each spin positioned at $\boldsymbol{r}$ as the average height of the three height vertices adjacent to the spin, as

$$\phi_{\boldsymbol{r}} = -\frac{1}{2} + \frac{1}{3}\sum_{i=1}^{3}\phi_{\boldsymbol{r}+\boldsymbol{e}_i}, \tag{D.1}$$

where we refer to $\phi_{\boldsymbol{r}}$ as the height of the spin positioned at $\boldsymbol{r}$. Next, we note that in each tower of prism tiles there is exactly one 3-arrow tile $Y_4$, $Y_5$ or $Y_6$ tile, with $Y = R, L$. Below these tiles, only one appropriate 2-arrow tile $Y_1$, $Y_2$ or $Y_3$ can be placed. Above the 3-arrow tiles, any of the 4-arrow tiles $Y_7$, $Y_8$ or $Y_9$ can be placed. This is similar to the case of the cubical tiles for the 6-vertex model, letting the prisms with $n$ arrows correspond to the cubical tiles with $n$ arrows, for $n = 2, 3, 4$. The main difference is that there are three prism tiles with 4 arrows that can be placed above the prism tiles with 3 arrows, compared to just the one cubical tile with 4 arrows $X_3$. The level at which the prism tile with 3 arrows is placed in the tower of prism tiles at $\boldsymbol{r}$ is given in terms of the height $\phi_{\boldsymbol{r}}$, exactly as in Eq. (B.2)

$$h_{\boldsymbol{r}} = M - \phi_{\boldsymbol{r}}, \tag{D.2}$$

where $M$ is the maximal possible height of any spin in the system we consider. As in the previous section, we have that the number of tiles $M_{\boldsymbol{r}}$ in the tower of prism tiles at $\boldsymbol{r}$ is determined by the range of possible values of $h_{\boldsymbol{r}}$, which is larger in the middle of the system than near the boundaries. That is, the further away $\boldsymbol{r}$ is from the boundary, the larger $M_{\boldsymbol{r}}$ is. This again gives an inverse step pyramid like shape of the 3D valid tiling. To see that arrowed paths in the valid tilings always obey Eq. (D.2), we consider the individual Dyck walks present in the lozenge tilings in Fig. (2). There are two main differences between the individual Dyck walks in the lozenge tilings and in the spin configurations of the 6-vertex model. Firstly, the "spins" in the lozenge tilings are half lozenges and each lozenge participates in two Dyck walks. Since we have allocated arrows due to both Dyck walks in each prism tile corresponding to a half lozenge, cutouts of the valid tilings in Fig. (2) does not exactly correspond to single Fredkin tilings as the cutouts in Fig. (7) did. Secondly, in the lozenge tiling model the set of bulk "spins" in a Dyck walk between two boundary spins changes depending on which lozenge tiling we consider. This can be seen by considering the lozenge tilings corresponding to the minimum and maximum height tilings seen in panel (a) in Fig. (14). Clearly, the bulk spins in a Dyck walk between the same boundary spins are different in the two figures. However, each lozenge tiling defines a Dyck walk between two boundary spins including a particular set of bulk spins. By considering this set of spins, bulk spins and boundary spins, as a spin chain, the arguments of Sec. B.1 will work in the same manner here. In particular, one notes that the height of a spin $\phi_{\boldsymbol{r}}$ is equal to the number of unpaired up steps behind the spin in either of the two Dyck walks, each being a source of an arrowed path arching above the spin at $\boldsymbol{r}$. Since none of the tiles are without arrows, or contains sources or drains of arrows, the arrowed paths have to turn at the maximal height $h_{\boldsymbol{r}}$ for a tiling to be valid. This the same reasoning leading to Eq. (B.2) in Sec. B.1 and the argument showing that it is impossible to tile a valid tiling for configurations not described by Dyck walks, works in the same way here. But due to the three different 4-arrowed prism tiles, we need to be extra careful to ensure that there is only one valid tiling for each lozenge tiling described by Dyck walks. We will show this in Sec. D.2.

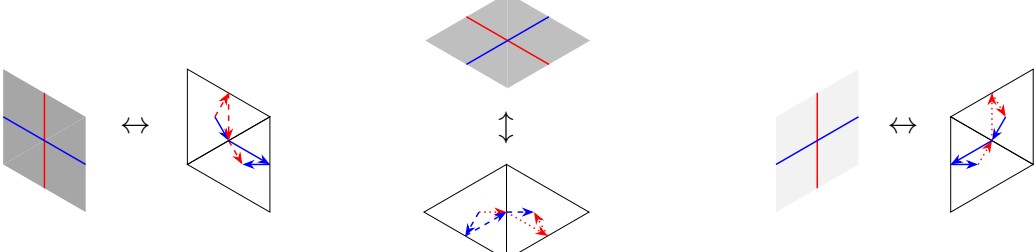

Figure 27: Each lozenge and its corresponding valid tiling in terms of the prism tiles at the level $l = h_{\boldsymbol{r}}$, where $h_{\boldsymbol{r}}$ is given for each triangle in Eq. (D.2). Below these tiles, the appropriate two arrowed tile will be placed and above these tiles any of the four arrowed tiles will be placed.

To show that any valid tiling of the prism tiles correspond to a lozenge tiling of the three lozenges in Fig. (2), we consider the towers of $R$ and $L$ prism tiles corresponding to the two halves of a lozenge. The height at the two lozenge halves, or "spins", are the same, as can be verified by considering the height vertices around a lozenge and using Eq. (D.1). The three-arrowed tiles in the two towers will therefore appear at the same level $l$ in the tensor network, as dictated by Eq. (D.2). In Fig. (27), the prism tiles at level $l$ are shown for the various lozenges. It is understood that at lower levels beneath the three arrowed tiles only the appropriate 2-arrowed tile can be placed. For example, beneath the $R_4$ tile, we have a tower of $R_1$ tiles as it is the only matching tile. Above the three arrowed tiles, any of the $Y_7 - Y_9$ tiles can be placed. Note that for each of the prism tiles in Fig. (27), there is no other tile matching at the edge joining the prism tiles than the ones indicated. This ensures that each valid tiling of the prism tiles correspond to a lozenge tiling. If two arrows of the same type had been allocated to each lozenge half, instead of two arrows of different type, the resulting $Y_4 - Y_6$ tiles would not have had one edge matching only one other tile. Within the tile framework presented here, this would open up for valid tilings of the prism tiles not corresponding to a lozenge tiling.

## D.2   Surjectivity

Since there are three tiles with 4 arrows in both sublattices, $Y_7$, $Y_8$ and $Y_9$, which fit above the tiles with 3 arrows, we need to be extra careful when arguing for the one-to-one correspondence between valid tilings of prisms and lozenge tilings $|T^C\rangle$. We now show that there is only one way to place the 4-arrow tiles for a given lozenge tiling. First, we note that each 4-arrow tile has two edges where three arrows meet. Each of these edges match only with one other 4-arrow tile. For example, the $\boldsymbol{k}_3$ (resp. $\boldsymbol{k}_2$) side of $R_7$ matches only $L_7$ (resp. $L_9$). Therefore, 4-arrow tiles must always appear together in groups of 6 as a single hexagonal configuration, in order to give a valid tiling. This is seen in the $l = 2$ level in Fig. (14), where the hexagon of 4-arrow tiles is marked with thick lines. Next, we observe that at each level $l$ in the tensor network, all tiles that are in a tower of tiles corresponding to a spin of height $\phi_{\boldsymbol{r}} > M - l$ will be 4-arrow tiles, where $M$ is the highest level of the tensor network. Now we show that clusters of spins of height $\phi_{\boldsymbol{r}} > M - l$ always form domains that can be tiled in exactly one way by hexagons. First, define the domain $\mathcal{R}_l$ to contain all vertices $u$ that is adjacent to the spin at $\boldsymbol{r}$, for which $\phi_{\boldsymbol{r}} > M - l$. Mathematically, the domain $\mathcal{R}_l$ is defined as

$$\mathcal{R}_l = \{u \in \mathcal{R} \mid u \text{ is located at } \boldsymbol{r} + \boldsymbol{e}_i, \ \phi_{\boldsymbol{r}} \geq M - l, \ i \in \{1, 2, 3\}\}, \qquad \text{(D.3)}$$

where $\boldsymbol{e}_i$ (for $i = 1, 2, 3$) are the displacement vectors from a spin at $\boldsymbol{r}$ to the height vertices at $\boldsymbol{r} + \boldsymbol{e}_i$ and $\mathcal{R}$ is the domain which contains all height vertices $u$. Note that $\mathcal{R}_l$ is not

necessarily simply connected. It might contain holes or consist of several disconnected domains, depending on the height profile of the given lozenge tiling. By definition, the boundary of $\mathcal{R}_l$ is at constant height $M - l$ for all $1 \leq l \leq M$, or more precisely, height vertices along the boundary alternates between height $M - l \pm 1/2$. A special case of this is seen in Fig. (2), where all spins satisfy $\phi_{\boldsymbol{r}} \geq 0$, the boundary edges have height 0 and the boundary height vertices have alternating heights $\pm 1/2$. This property of the height at the boundary vertices can be written formally as

$$\forall u, v \in \partial \mathcal{R}_l : |\phi_u - \phi_v| \leq 1, \tag{D.4}$$

where $u$ and $v$ are height vertices and $\partial \mathcal{R}_l$ is the boundary of $\mathcal{R}_l$. This is a stronger constraint than the requirement that a domain is tileable by lozenges, seen in Eq. (2.7). We note that Eq. (D.4) implies that adjacent edges on the boundary of a domain $\mathcal{R}_l$ must be rotated 120° relative to each other, as follows from the height convention defined in Fig. (2). Therefore for a domain $\mathcal{R}_l$ satisfying Eq. (D.4), we have that any segment of its boundary can be described by connecting a set of the dashed edges of the hexagon in panel (a) in Fig. (28). Two examples of such segments can be seen in panel (b) and (c) of Fig. (28). Note that the height vertex $c$ is always in the bulk $\mathcal{R}_l^\circ$.

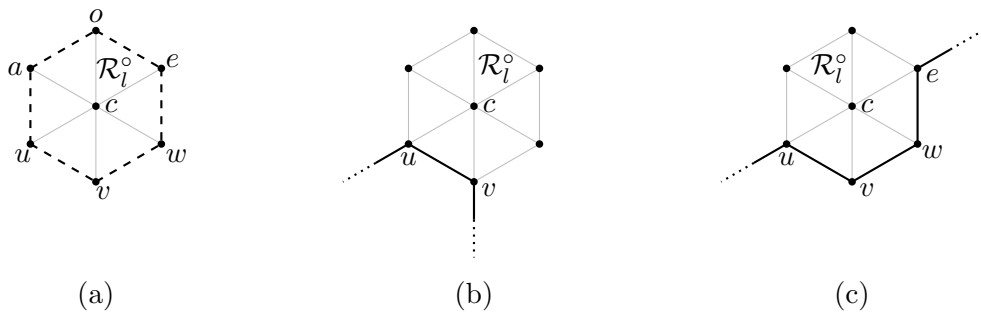

(a)  (b)  (c)

Figure 28: (a) Any segment of any boundary of a domain $\mathcal{R}_l$, satisfying Eq. (D.4), can be constructed by connecting a set of dashed lines in the figure. (b)-(c) Two boundary segments, where a set of dashed edges in (a) have been connected. The labeled height vertices (except $c$) indicate height vertices on the boundary and the solid black lines connecting them are edges on the boundary. The unfinished black lines indicate the boundary edges outside the boundary segment and the light gray lines indicate edges in the bulk $\mathcal{R}_l^\circ$.

In Fig. (28) (b), we take only the edge $u - v$ to be on the boundary, and the rest to be in the bulk. In Fig. (28) (c), the edges $u - v$, $v - w$ and $w - e$ are on the boundary. This defines two boundary segments that can be present in a boundary $\partial \mathcal{R}_l$ satisfying Eq. (D.4). Note that the vertex $c$ can not be on the boundary, as that would violate Eq. (D.4), and each of the boundary segments therefore have a corresponding bulk vertex $c$. We now wish to remove a hexagonal unit of area containing the height vertex $c$ in the bulk, to produce a new domain $\mathcal{R}_l'$. To make sure that the new domain $\mathcal{R}_l'$ also satisfies Eq. (D.4), and therefore have a boundary that can be described by Fig. (28), the hexagonal area containing $c$ can only be removed in one way for each different boundary segment. This is achieved by removing $c$ and the height vertices on the boundary that only are connected to other height vertices on the boundary and to $c$. Examples of this, can be seen for the two different boundary segments in Fig. (29)

Since the boundary of $\mathcal{R}_l'$ also satisfies Eq. (D.4), each segment of its boundary is also described by Fig. (28), and have a corresponding bulk vertex $c$. Therefore, by continuing the process of removing height vertices in the same way as done above, each step produces

Figure 29: Panel (a) (resp. panel (b)) shows the only way to remove the hexagonal area containing the height vertex $c$ from the domain $\mathcal{R}_l$ for the boundary segment in Fig. (28) (b) (resp. (c)), that ensures that the boundary of the domain $\mathcal{R}'_l$ also satisfy Eq. (D.4).

a new domain with an area less than the previous domain by exactly one hexagon, and with a boundary that satisfies Eq. (D.4). This process of removing the height vertices can be continued until there is a single hexagon left, and therefore shows the only way of tiling the domain $\mathcal{R}_l$ by hexagons. We therefore conclude that a domain that satisfies Eq. (D.4) can be tiled in exactly one way by hexagons, which implies that domains $\mathcal{R}_l$ defined in Eq. (D.3) always can be tiled in exactly one way by hexagons. This in turn implies that at each level $l$ in the tensor network, the 4-arrowed tiles can always be tiled together in hexagons in exactly one way, concluding the proof that there is a one-to-one mapping between valid tilings of prism tiles and lozenge tilings. Now, we can argue for why we have chosen two arrows per spin. Consider the hexagon of 4-arrowed tiles in the $l = 2$ level of panel (c) in Fig. (14). If only one arrow per spin was chosen, only one of the two arrowed paths of the same arrow type would have been present. But this would have opened up for two different valid tilings for the same lozenge tiling, one tiling per path. By using two arrows per spin, we treat the paths symmetrically and include them both in the tiling so that only one tiling is possible per lozenge tiling.

# E  Determination of weighting for the 5-leg tensors

To determine the weigths of $q$ for the various prism tiles, seen in Table. 2, we will follow a different approach than we used for the 6-vertex model in Appendix C. As mentioned in Appendix D.1, the set of bulk spins taking part of a Dyck walk between two boundary spins, changes for different lozenge tilings. This prohibits using the same method as in Appendix C, as it depended on writing the height operator on the form of Eq. (C.2). Instead, we use the fact that there is only one 3-arrowed prism tile in each tower of tiles, with a tower of 2-arrowed (resp. 4-arrowed) tiles beneath (resp. above) it. This can be stated as the following equation

$$N_2(\boldsymbol{r}) = (M_{\boldsymbol{r}} - \phi_{\boldsymbol{r}} - 1), \quad N_3(\boldsymbol{r}) = 1, \quad N_4(\boldsymbol{r}) = \phi_{\boldsymbol{r}}, \tag{E.1}$$

where $N_a(\boldsymbol{r})$ is the number of tiles with $a$ arrows in the tower of tiles at $\boldsymbol{r}$, $\phi_{\boldsymbol{r}}$ is the height of the spin at $\boldsymbol{r}$ and $M_{\boldsymbol{r}}$ is the number of levels in the tower of prism tiles corresponding to the spin at $\boldsymbol{r}$. As we should, we have $N_2(\boldsymbol{r}) + N_3(\boldsymbol{r}) + N_4(\boldsymbol{r}) = M_{\boldsymbol{r}}$. That is, the number of tiles in the tower equals the number of levels in the tower. To find the appropriate values of the tiles we write the volume of a lozenge tiling $T$ as $V(T) = \sum_u \phi_u$, that is as a sum over the height at the height vertices $u$. The lozenge tiling $T$ will then be weighted by a factor of $q^{V(T)} = \prod_u q^{\phi_u}$. Firstly, we focus on spins in the bulk of our system. In the bulk, we let the factors of $q$ due to each height vertex $u$, $q^{\phi_u}$, be distributed evenly to the

six adjacent tower of tiles. This is the same as saying that the factor of $q$ that each tower of tiles must carry, $F_{\boldsymbol{r}}^{\text{tower}}(q)$ is given by

$$F_{\boldsymbol{r}}^{\text{tower}}(q) = q^{\frac{1}{6}(\phi_{\boldsymbol{r}+\boldsymbol{e}_1}+\phi_{\boldsymbol{r}+\boldsymbol{e}_2}+\phi_{\boldsymbol{r}+\boldsymbol{e}_3})} = q^{\frac{\phi_{\boldsymbol{r}}}{2}+\frac{1}{4}}, \tag{E.2}$$

where the $\phi_{\boldsymbol{r}+\boldsymbol{e}_i}$ for $i = 1, 2, 3$ is the height at the three adjacent height vertices to the tower of tiles corresponding to the spin at $\boldsymbol{r}$. The second equality in Eq. (E.2) follows by Eq. (D.1), where $\phi_{\boldsymbol{r}}$ is the height of the spin at $\boldsymbol{r}$. Now, we assume that the value of a tile only depends on the number of arrows in the tile, as the case was for the 6-vertex model. Then, we use that the number of tiles $N_a(\boldsymbol{r})$ with $a$ arrows in it, is given by Eq. (E.1), so that for the tower of tiles at $\boldsymbol{r}$ we have the following equation

$$F_{\boldsymbol{r}}^{\text{tower}}(q) = q^{N_2(\boldsymbol{r})x_2} q^{N_3(\boldsymbol{r})x_3} q^{N_4(\boldsymbol{r})x_4} \tag{E.3}$$

where $q^{x_i}$ is the value of a tile with $i$ arrows. Inserting Eq. (E.2) into Eq. (E.3), gives

$$(M_{\boldsymbol{r}} - \phi_{\boldsymbol{r}} - 1)x_2 + x_3 + \phi_{\boldsymbol{r}}x_4 = \frac{1}{2}\phi_{\boldsymbol{r}} + \frac{1}{4}. \tag{E.4}$$

The number of tiles in a tower of tiles at $\boldsymbol{r}$, $M_{\boldsymbol{r}}$, is only dependent on how far away $\boldsymbol{r}$ is from the boundary, as was established in Appendix D.1. We also have that $\phi_{\boldsymbol{r}} \in [0, M_{\boldsymbol{r}} - 1]$. For a given $\boldsymbol{r}$, Eq. (E.4) therefore gives a set of $M_{\boldsymbol{r}}$ equations, one for each possible value of $\phi_{\boldsymbol{r}}$. The solution to this set of equations is

$$x_3 = \frac{1}{4} - (M_{\boldsymbol{r}} - 1)x_2, \quad x_4 = \frac{1}{2} + x_2. \tag{E.5}$$

Since $M_{\boldsymbol{r}}$ depends on $\boldsymbol{r}$, and we want our tiles to take on the same values for all $\boldsymbol{r}$ in bulk, we choose $x_2 = 0$, which gives the values for the tiles seen in Table. 2. This matches the values obtained for the corresponding tiles in the 6-vertex model. This makes sense, as ground states of both models are described by Dyck walks.

To get the correct weighing of states, we must treat the factor $q^{\phi_u}$ due to height vertices $u$ on the boundary with extra care. This is because the height vertices on the boundary does not have six adjacent spins, as can be seen by considering the lozenge tilings in Fig. (2). Therefore, if we allocate the factors of $q$ as in Eq. (E.2), for all $\boldsymbol{r}$ we get a wrong weighing of the state, since we will miss part of the $q^{\phi_u}$-factor from height vertices $u$ on the boundary. To remedy this, we note that the height vertices $u$ on the boundary have either 2 or 4 adjacent spins, due to the constraint Eq. (D.4) for our domain $\mathcal{R}$. Examples of boundary vertices with only 2 adjacent spins are $v$ and $w$ in Fig. (28) (c), while the vertices $u$ and $e$ are examples of boundary vertices with 4 adjacent spins. For the spins on the boundary we know that the corresponding tower of tiles only have one level, and that the tile in that level has three arrows. Note that the height vertices on the boundary with 4 adjacent spins always have height $\phi_u = 1/2$, due to the constraint Eq. (D.4). Then, instead of using Eq. (E.2) to allocate the factors of $q$, we use

$$\begin{aligned}
F_{\boldsymbol{r}}^{\text{tower}}(q) &= q^{x_3'} = q^{\frac{1}{2}(\phi_1+\phi_2)+\frac{\phi_3}{6}}, \\
F_{\boldsymbol{r}}^{\text{tower}}(q) &= q^{x_3''} = q^{\frac{\phi_1}{2}+\frac{\phi_2}{3}+\frac{\phi_3}{6}},
\end{aligned} \tag{E.6}$$

for $\boldsymbol{r} \in \partial\mathcal{R}$. In Eq. (E.6), the vertices denoted by 1 and 2 lie on the boundary while the vertex denoted by 3 is in the bulk. The upper equation, is used when the spin at $\boldsymbol{r}$ has two adjacent height vertices on the boundary with only 2 adjacent spins, so that we have put the square root of the $q$ factor due to both spins 1 and 2 into the the tower of tiles

at $r$. The lower equation is used when the height vertex 2 has 4 adjacent spins. By using Eq. (E.6) we include all factors of $q$ from the height vertices. The factors $q^{x'_3}$ and $q^{x''_3}$ denotes the appropriate value of the three arrowed tiles on the boundary. Due to the fact that the lozenges are fixed on the boundary, the height at the height vertex 3, which is in the bulk, is always $\phi_3 = 3/2$. This can be used to simplify Eq. (E.6). Furthermore, we have that $\phi_1 = -\phi_2$, in the upper equation of Eq. (E.6), due to the constraint that boundary spins must alternate in height. Also, in the lower equation of Eq. (E.6), we know that $\phi_1 = -\phi_2 = -1/2$, since the height vertex 2 is the height vertex with 4 adjacent spins. Using this we can solve for $x'_3$ and $x''_3$, which gives

$$x'_3 = \frac{1}{4}, \ \ x''_3 = \frac{1}{6}. \tag{E.7}$$

Therefore, the the value of the three arrowed tiles on the boundary, that has one adjacent height vertex with 4 adjacent spins, must be $1/6$ instead of $1/4$. This must be incorporated in the boundary tensor (on the "wall indices) we contract with in Fig. (16), which takes on the value of $1/6 - 1/4 = -1/12$ at these sites. This is seen as the weighted boundary tensor $q^{-\frac{1}{12}} \delta_{\boldsymbol{k}_i,(\boldsymbol{0},\boldsymbol{0},\boldsymbol{0})}$ in Fig. (16).

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
