# Peer review of "Highly Entangled 2D Ground States: Tensor Networks and Correlation Functions"

_SciPost Physics_

## Round 1 · Referee Report · Anonymous (Referee 1) · 2025-8-3

Strengths

1- Non trivial results 2- novel construction of a 3d exact tensor network describing the ground state of a 2D quantum model 3- The paper supplies clear motivations and background material

Weaknesses

1- Results are of academic interest 2- Require significant effort to fully understand.

Report

In this paper, the authors construct an exact tensor network description of a class of 2D height models that serve as paradigms for exotic phase transitions, particularly in terms of entanglement. These models are frustration-free and allow our understanding of the ground state as a geometrical object. The construction is challenging in that it involves constructing 3D surfaces and the 3D tensor network, and the authors appear to have succeeded in doing so, as far as I could verify. The main results are the descriptions of the 3D tensors, as depicted in Figures 5 and 12. The authors also provide some insight into correlation functions, which are summarized nicely in tables at the end, both for height and color correlations. The appropriate literature is cited in a transparent and helpful manner, and an extensive appendix covers both the original setup (the definition of the models is lengthy by itself) as well as many details of the main text. Clearly, much work has gone into writing this paper and, in particular, polishing the presentation.
The results serve to complete an obvious next step in the field of highly entangled states - that of an explicit tensor network in 3D, and as such are of sufficient quality and interest for publication.
The only thing I would suggest that can improve the paper, but this is optional (I would at least suggest that the authors do that if they give talks on the subject!) is to find a way to illustrate the actual 3D construction in a more intuitive way than just listing the prisms that do the job.

Recommendation

Publish (easily meets expectations and criteria for this Journal; among top 50%)

---

## Round 1 · Referee Report · Anonymous (Referee 2) · 2025-9-15

Strengths

The manuscript addresses the problem of constructing exact tensor network representations for highly entangled ground states in two dimensions, extending the holographic tensor networks known for 1D Fredkin and Motzkin chains. These models are among the rare examples that exhibit transitions between area-law and extensively scaling entanglement entropy. Given the general lack of analytically tractable examples of such exotic entanglement structures in higher dimensions, the present work makes a valuable contribution by formulating explicit tensor networks based on cubic and prismatic tilings. The technical development is sound, and the presentation is detailed. The results will be of interest to researchers working on tensor networks, entanglement scaling, and related aspects of quantum many-body physics.

Report

The paper is already well written and technically sound. I only have two minor comments concerning the references:

On page 2, the authors mention the entanglement entropy of "free fermionic systems with a Fermi sea in any dimension," citing Ref. [9]. I suggest replacing "Fermi sea" with "Fermi surface": the term "Fermi sea" could also refer to a gapped situation (e.g., a fully occupied band separated by an empty band with a gap) or systems with only Dirac points, where the area law is restored without a logarithmic violation. In addition, the parallel work by M. Wolf, Phys. Rev. Lett. 96, 010404 (2006), published back-to-back with Ref. [9], should also be cited. Both papers independently established the logarithmic violation of the area law for free fermions with a Fermi surface.

Regarding the higher-dimensional generalization related to Ref. [10], the authors currently cite Ref. [11] only. However, an earlier contribution by G. Sierra and collaborators, J. Stat. Mech. (2015) P06002, should also be cited for completeness.

Other than these points, I have no major criticisms. The technical constructions in the main sections appear correct and carefully explained.

In conclusion, I recommend publication in SciPost Physics once the above citation issues are addressed.

Recommendation

Publish (easily meets expectations and criteria for this Journal; among top 50%)

---

## Round 1 · Referee Report · Anonymous (Referee 3) · 2025-12-5

Strengths

1- The work presented exact construction of the 3D tensor network representation of the ground state of 2D quantum models with volume entanglement.

2- The mapping between the tiling and tensor network construction is quite intriguing.

Weaknesses

1- The physical significance seems to be not completely clear, such as the connection to holography.

2- The paper is written in a quite technical manner, making it difficult for readers not in the field to comprehend.

Report

The paper studied the exact tensor network representations of 2D ground states of quantum lattice models with transitions between area-law and volume-law entanglement. The generalisation to the 1D counterpart, i.e. multi-colour Fredkin chain is studied extensively.

The tensor network part seems to be technically sound and explicit, but it is still quite difficult to study.

The correlation function part is a bit off the main topic of the rest of the paper. It used different techniques, such as mapping the correlation functions in terms of the random-height fluctuations. I don't see the connection to the tensor-network construction in the previous sections, and I think that this needs to be justified, which I do not see that the mapping holds immediately. I would suggest the authors to explain if there's any connection between the correlation functions and the exact tensor-network wavefunction, as well as some details on why the mapping works. It seems that at $q=1$ (transition point), the correlation functions should be related to Gaussian free field. I wonder if this is related to the results.

Recommendation

Publish (meets expectations and criteria for this Journal)

---

## Editorial Decision

in_refereeing